# A single N-terminal amino acid determines the distinct roles of histones H3 and H3.3 in the *Drosophila* male germline stem cell lineage

Chinmayi Chandrasekhara[1]ᐤ, Rajesh Ranjan[1,2]ᐤ, Jennifer A. Urban[1]ᐤ, Brendon E. M. Davis[1], Wai Lim Ku[3], Jonathan Snedeker[1], Keji Zhao[3], Xin Chen[1,2]*

1 Department of Biology, The Johns Hopkins University, Baltimore, Baltimore, Maryland, United States of America, 2 Howard Hughes Medical Institute, Department of Biology, The Johns Hopkins University, Baltimore, Maryland, United States of America, 3 Systems Biology Center, National Heart, Lung and Blood Institute, NIH, Bethesda, Maryland, United States of America

ᐤ These authors contributed equally to this work.
* xchen32@jhu.edu

**Data Availability Statement:** All relevant data are within the paper and its Supporting Information files. In addition, the ChIC-seq data have been

## Abstract

Adult stem cells undergo asymmetric cell divisions to produce 2 daughter cells with distinct cell fates: one capable of self-renewal and the other committed for differentiation. Misregulation of this delicate balance can lead to cancer and tissue degeneration. During asymmetric division of *Drosophila* male germline stem cells (GSCs), preexisting (old) and newly synthesized histone H3 are differentially segregated, whereas old and new histone variant H3.3 are more equally inherited. However, what underlies these distinct inheritance patterns remains unknown. Here, we report that the N-terminal tails of H3 and H3.3 are critical for their inheritance patterns, as well as GSC maintenance and proper differentiation. H3 and H3.3 differ at the 31st position in their N-termini with Alanine for H3 and Serine for H3.3. By swapping these 2 amino acids, we generated 2 mutant histones (i.e., H3A31S and H3.3S31A). Upon expressing them in the early-stage germline, we identified opposing phenotypes: overpopulation of early-stage germ cells in the H3A31S-expressing testes and significant germ cell loss in testes expressing the H3.3S31A. Asymmetric H3 inheritance is disrupted in the H3A31S-expressing GSCs, due to misincorporation of old histones between sister chromatids during DNA replication. Furthermore, H3.3S31A mutation accelerates old histone turnover in the GSCs. Finally, using a modified Chromatin Immunocleavage assay on early-stage germ cells, we found that H3A31S has enhanced occupancy at promoters and transcription starting sites compared with H3, while H3.3S31A is more enriched at transcriptionally silent intergenic regions compared to H3.3. Overall, these results suggest that the 31st amino acids for both H3 and H3.3 are critical for their proper genomic occupancy and function. Together, our findings indicate a critical role for the different amino acid composition of the N-terminal tails between H3 and H3.3 in an endogenous stem cell lineage and provide insights into the importance of proper histone inheritance in specifying cell fates and regulating cellular differentiation.

deposited. GEO accession number for the ChIC-seq data is GSE212936.

**Funding:** This work was supported by the National Institutes of Health (F32 GM134664 to C.C., K99 GM145973 to J.A.U., 5T32GM007231 to B.D. and J.S., F31 HD104526 to J.S., and R35GM127075 to X.C.), as well as Division of Intramural Research, NHLBI to K.Z.), American Cancer Society grant# 133950-PF-19-131-01-DMC to J.A.U, the Howard Hughes Medical Institute, the David and Lucile Packard Foundation, and Johns Hopkins University startup funds to X.C. The funders had no role in study design, data collection and analysis, decision to publish, or preparation of the manuscript.

**Competing interests:** The authors have declared that no competing interests exist.

**Abbreviations:** ACD, asymmetric cell division; ChIC, chromatin immunocleavage; CySC, cyst stem cell; FB, fetal bovine serum; GB, gonialblast; GSC, germline stem cell; PBS, phosphate-buffered saline; PTM, posttranslational modification; SEM, standard error of the mean; SG, spermatogonial cell; SRCF, superresolution imaging of chromatin fibers; TSS, transcription start site; WT, wild-type.

## Introduction

Adult stem cells have the unique capability of self-renewal and the ability to differentiate. This balance could be achieved by asymmetric cell division (ACD), which gives rise to 2 daughter cells with distinct fates, one daughter cell with the ability to self-renew and the other daughter cell that is committed to differentiation. ACD occurs widely during development as well as tissue homeostasis and regeneration [1–5]. Imbalance between self-renewal versus differentiation of adult stem cells can result in cancer, tissue degeneration, infertility, as well as aging [6–11].

In eukaryotes, epigenetic mechanisms play a critical role in defining cell identities and functions. There are 2 types of histone proteins: Canonical nucleosomal core histones are mainly incorporated during DNA replication (i.e., H3, H4, H2A, and H2B) [12]; histone variants are incorporated in a replication-independent manner [13]. Epigenetic mechanisms such as DNA methylation, posttranslational modifications (PTMs) of histones, histone variants, and chromatin remodeling can instruct cells with identical genomes to turn on different sets of genes and take on distinct identities [14–18]. However, in multicellular organisms, how epigenetic information is maintained or changed during cell divisions, in particular through ACDs, to give rise to daughter cells with distinct cellular fates remained largely unclear [19–23].

*Drosophila melanogaster* gametogenesis represents an ideal model system to study mechanisms regulating the maintenance and proliferation of adult stem cells, as well as proper differentiation of stem and progenitor cells [24, 25]. In *Drosophila*, both male and female germline stem cells (GSCs) can undergo ACD to give rise to self-renewed stem daughter cells and the other daughter cells, which give rise to mature gametes upon differentiation. Male GSCs attach to a group of postmitotic somatic cells called hub cells. A male GSC divides asymmetrically to give rise to both a self-renewed GSC and a gonialblast (GB), the daughter cell that initiates proliferation followed by meiosis and terminal differentiation to become sperm. GBs first go through a transit-amplifying stage with 4 rounds of mitosis as spermatogonial cells (SGs). Once spermatogonial proliferation is complete, cells enter the spermatocyte stage when they initiate a robust gene expression program and epigenomic changes to prepare for meiotic divisions and spermatid differentiation [25–28] (Fig 1A).

Previously, it has been reported that during ACD of *Drosophila* male GSC, preexisting (old) canonical histones H3 and H4 are selectively inherited by the GSC, whereas newly synthesized (new) H3 and H4 are enriched in the differentiating daughter GB cell [29,30]. Intriguingly, this phenomenon is unique to the canonical histones H3 and H4. By contrast, histone variant H3.3 does not exhibit such a global asymmetric pattern [29]. While significant progress has been made in understanding how epigenetically distinct sister chromatids enriched with old versus new H3 are specifically established and recognized, our understanding of what underlies the observed differences between H3 and H3.3 inheritance patterns during ACD of GSCs is limited. Histone variants, such as H3.3, H2A.Z, and CENP-A, are involved in important biological functions, such as transcription, DNA repair, and defining centromeres [31–34]. Between H3 and H3.3, H3 is primarily incorporated during DNA replication, while H3.3 is incorporated in a replication-independent manner [33,35].

Recent studies have shown that point mutations at multiple residues of histones and histone variants result in dramatic cellular defects and human diseases [36,37]. Our previous work showed that mutations at the Threonine 3 residue of H3 [e.g., H3T3A (Thr3 to Ala) and H3T3D (Thr3 to Asp)] lead to *Drosophila* male GSC loss, progenitor germline tumors, and progressively decreased male fertility [38]. Mutations at critical histone and histone variant residues have also been identified in a variety of human cancers [36]. For example, mutation at the Lys27 residue of H3.3 [H3.3K27M (Lys27 to Met)] or H3.1 (H3K27M) have both been identified in pediatric glioblastoma [2,37,39]. Another critical residue is located at position 31

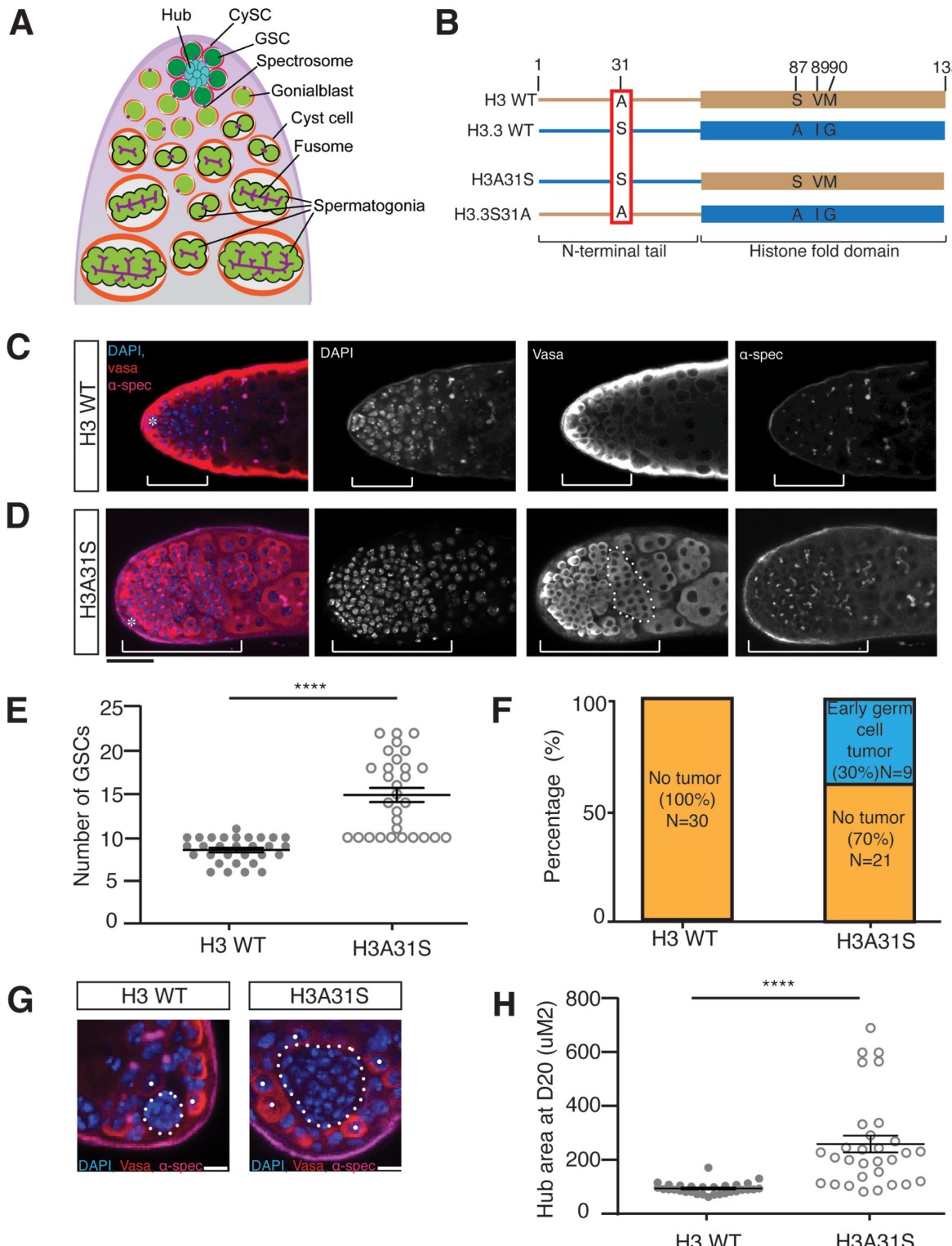

**Fig 1. Cellular defects in *nos-Gal4; UAS-H3A31S* testes.** (A) An illustration of the apical tip of the *Drosophila* testis. The hub (cyan) is a cluster of 10–12 densely packed somatic cells. The GSCs (dark green) and the CySCs (magenta) are radially positioned around the hub, with 2 CySCs enveloping each GSC. The GSCs undergo an asymmetric division to produce a self-renewed daughter GSC and a GB (light green). The GB subsequently leaves the hub and undergoes 4 rounds of miotic divisions with incomplete cytokinesis to create cysts of interconnected SGs. The differentiating germline cysts continue to be encapsulated by 2 postmitotic cyst cells (orange). Round

spectrosome (purple) is detected in early-stage germline including GSCs and GBs, which is branched as fusome (purple) in late-stage SGs. Adapted from [54]. (**B**) Diagram of the single amino acid differences at the N-terminal tails in WT H3 and H3.3 proteins. A point mutation in the N-terminal tail of H3 changing alanine to serine at position 31 is named H3A31S. A point mutation in the N-terminal tail of H3.3 changing serine to alanine at position 31 is named H3.3S31A. (**C, D**) Overpopulation of the early-stage germ cells in H3A31S-expressing testes compared to H3-expressing testes, both from males aged at 29 degrees for 20 days. The early-stage germ cells region is demonstrated by the brackets, and an overproliferative mitotic spermatogonial cyst is depicted by dotted outline. Immunostaining using anti-Vasa (red), DAPI (blue), anti-α-Spectrin (magenta). Asterisk: niche. Scale bars: 20 μm. (**E**) Quantification of GSCs in H3-expressing testes ($n = 30$, $8.6 \pm 0.3$) and H3A31S-expressing testes ($n = 30$, $14.9 \pm 0.8$). Unpaired $t$ test to compare the 2 individual datasets to each other. ****: $P < 0.0001$. Error bars represent the SEM. (**F**) Quantification of early germ cell tumors in H3- and H3A31S-expressing testes. H3 exhibited no tumors (ratio = 0/30) and 30% of H3A31S-expressing testes exhibited early germ cell tumors (ratio = 9/30). (**G**) Expanded hub area in H3A31S-expressing testis compared to H3-expressing testis. The hub areas are depicted by the dotted outline. Individual white dots indicate GSCs. Scale bars: 5 μm. (**H**) Quantification of the hub area of H3- and H3A31S-expressing testes. H3 ($n = 30$, $93.6 \pm 3.8\ \mu m^2$) and H3A31S ($n = 30$, $258.1 \pm 31.3\ \mu m^2$) expressing testes. Unpaired $t$ test to compare the 2 individual datasets to each other. ****: $P < 0.0001$. All data are Avg ± SEM. Error bars represent the SEM. $P < 0.0001$. The data underlying (**E, F, H**) can be found in S1 Table. CySC, cyst stem cell; GB, gonialblast; GSC, germline stem cell; SEM, standard error of the mean; SG, spermatogonial cell; WT, wild-type.

of H3 and H3.3. This is the only distinct amino acid between H3 and H3.3 at their N-termini (Fig 1B), which has been shown to be mutated in human ovarian cancer, squamous cell carcinomas, and colorectal adenocarcinomas samples [36]. To better examine the importance of this residue, we swapped this particular amino acid in both H3 and H3.3 to generate 2 hybrid mutant histone proteins (Fig 1B). We then expressed each of them in the *Drosophila* early-stage male germline to investigate whether the mutant histones lead to any germline defect. Our data demonstrate a spectrum of phenotypes, indicating that the 31st residue plays a vital role in regulating proper germline activities. Collectively, our studies provide in vivo insights into how the N-terminal protein composition differences between H3 and H3.3 dictate their roles in specifying distinct cell fates and their inheritance patterns during ACD of male GSCs.

## Results

### Expression of mutant H3A31S or H3.3S31A in early-stage germline causes opposing defects

H3 and H3.3 proteins are 97% identical at the primary sequence level (Fig 1B). Of the 4 distinct amino acids between H3 and H3.3, the role of the 3 amino acids toward the C-termini is well defined, as they allow for specific chaperone binding, nucleosome splitting behavior, and unique chromatin incorporation modes [35,40–42]. On the other hand, the N-terminal tails of histones interact with DNA to modulate changes at local chromatin landscape and thus gene expression, highlighting the importance of the histone tail sequences [16–18]. However, little is known regarding how the difference at position 31 between H3 and H3.3 may account for their distinct biological functions.

To examine the roles of the distinct 31st amino acid at the N-terminal tails of H3 and H3.3, we mutated this amino acid of both the canonical histone H3 and the variant histone H3.3. We mutated the alanine residue in H3's N-terminal tail to a serine (H3A31S) and the serine residue in the H3.3's N-terminal tail to an alanine (H3.3S31A) (Fig 1B). We then expressed these mutant histones labeled with fluorescent tags in early-stage germline including GSCs using the *nanos-Gal4* (*nos-Gal4*) driver [29,43]. Next, we analyzed the cellular defects in both *nos>H3A31S* and *nos>H3.3S31A* testes. Both wild-type (WT) H3- and H3.3-expressing testes, driven by the same *nos-Gal4* driver, were used as controls.

First, we found that early-germline expression of H3A31S results in overpopulation of early-stage germ cells, including GSCs, GBs, and mitotic SGs (Fig 1D), while the control WT H3-expressing testes did not display such a phenotype (Fig 1C). Additionally, the dotted spectrosome structure is unique in GSCs and GBs but becomes a branched fusome structure in more differentiated SGs and spermatocytes. Using the morphologic differences of nuclei and

the spectrosome versus fusome, staging of the germline can be distinguised [44–49]. There was an expansion of cells with the round spectrosome morphology and condensed nuclei at the tip of the *nos>H3A31S* testes (Fig 1D), compared to the control testes (Fig 1C). Quantification of this phenotype indicated that 30% of the *nos>H3A31S* testes (Fig 1D and 1F) had an early germ cell overpopulation phenotype compared to no such phenotype in WT H3-expressing testes (Fig 1C and 1F). Because we only classified testes with obviously expanded progenitor germ cell zones as the ones with early germline tumor phenotype, this ratio (30%) reflects an incomplete penetrance of this phenotype when evaluated under the current experimental conditions. In addition, the average number of GSCs was 14.9 in H3A31S-expressing testes, which is significantly more than 8.6, the average number of GSCs in WT H3-expressing testes (Fig 1E), consistent with previous reports of GSC number in testes from WT fly strains [50–53]. Finally, we observed a significant increase in the hub area in H3A31S-expressing testes compared to WT H3-expressing testes (Fig 1G and 1H). This is likely a secondary defect due to GSC defects, as reported previously [52,54]. Collectively, these data indicate that the 31st amino acid in the H3 tail is important for GSC identity and niche architecture, as well as for proper differentiation of early-stage germ cells, such as GSCs, GBs, and SGs.

Next, we examined the potential phenotypes in the H3.3S31A-expressing testes. We observed a gradual loss of GSCs in the H3.3S31A-expressing testes, compared to the WT H3.3-expressing testes in adulthood during aging (Fig 2A). A significant decrease in the number of GSCs was detected in adult testes from 5-day- to 10-day-old males, even though the GSCs in both genotypes showed comparable number in testes from 1-day-old males, suggesting that the GSC loss is due to maintenance but not establishment defects (Fig 2B). Consistently, the male fertility between *nos>WT H3.3* and *nos>H3.3S31A* males was comparable in young adults. However, *nos>H3.3S31A* males showed more significantly declined fertility over time than the control *nos>H3.3* males (Fig 2C). Age-dependent decrease of GSC activity and male fertility has been reported previously [38,55–58]. Notably, there was an approximate 10-day delay for the decreased male fertility phenotype compared to the GSC loss phenotype (Fig 2C versus Fig 2B), in line with the time needed for spermatogenesis from GSC to mature sperm, which takes approximately 10 days at room temperature [59,60]. These results suggest that the GSC loss contributes to the decreased male fertility phenotype in *nos>H3.3S31A* males.

To further investigate whether GSCs fail to self-renew in H3.3S31A-expressing testes, we examined the levels of Stat92E, a transcription factor that is required for GSC identity and maintenance [44,50,53,61–63]. Indeed, using parallel data acquisition and analyses for all GSCs across different genetic backgrounds, we detected decreased levels of Stat92E immunostaining signals were detected in H3.3S31A-expressing GSCs compared to WT H3.3-expressing GSCs (Fig 2D and 2E). Together, these results suggest that the 31st amino acid in the N-terminal tail of H3.3 is required for GSC maintenance.

## H3A31S and H3.3S31A show globally symmetric segregation patterns during ACD of *Drosophila* male GSCs

Previously, expression of mutant H3 with point mutations at the Thr 3 residue within the N-terminus (e.g., H3T3A and H3T3D) results in randomized histone inheritance patterns, as well as phenotypes including male GSC loss, progenitor germline tumors, and progressively decreased male fertility [38]. Given the similar cellular defects discovered for GSCs expressing either H3A31S or H3.3S31A, we next investigated whether these mutations on the 31st residue on WT H3 and H3.3 affect their inheritance patterns during ACD of male GSCs. To examine mutant histone inheritance patterns during GSC ACDs, we used a two-color system to

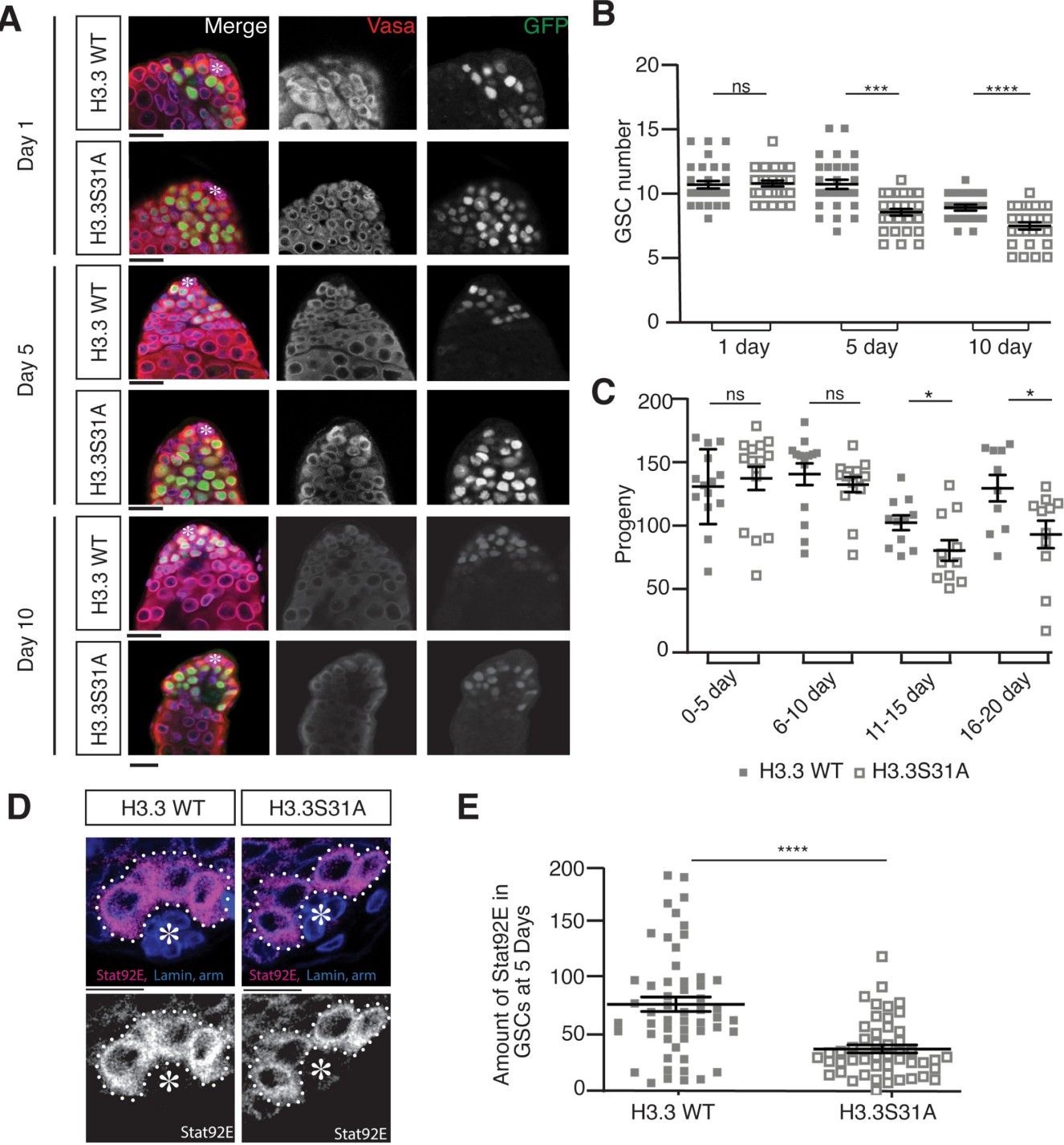

**Fig 2. Cellular defects in *nos-Gal4; UAS-H3.3S31A* testes.** (**A**) Immunostaining of *nos-Gal4; UAS-H3.3-GFP* and *nos-Gal4; UAS-H3.3S31A-GFP* testes showing decreasing germ cells during aging. Immunostaining using anti-Vasa (red), DAPI (blue), anti-α-Spectrin, and anti-Armadillo (magenta). Asterisk: niche. Scale bars: 15 μm. (**B**) Quantification of GSCs in H3.3- and H3.3S31A-expressing testes showing decreasing GSCs during aging. Day 1: H3.3 ($n = 30$, 10.6 ± 0.3) and H3.3S31A ($n = 30$, 10.7 ± 0.2); Day 5: H3.3 ($n = 30$, 10.7 ± 0.4) and H3.3S31A ($n = 30$, 8.5 ± 0.3); Day 10: H3.3 ($n = 22$, 8.9 ± 0.2) and H3.3S31A ($n = 27$, 7.4 ± 0.3). All data are Avg ± SEM. Unpaired *t* test to compare the 2 individual datasets to each other. ****: $P < 0.0001$.***: $P = 0.0005$. ns, not significant. Error bars represent the SEM. (**C**) Quantification of progenies from crosses using males expressing either WT H3.3 or H3.3S31A in the testes, which show decreasing fertility over time. Day 0–5: H3.3 ($n = 15$, 129.0 ± 7.8) and H3.3S31A ($n = 15$, 135.4 ± 9.1); Day 6–10: H3.3 ($n = 14$, 138.7 ± 8.4) and H3.3S31A ($n = 14$, 130.6 ± 5.9); Day 11–15: H3.3 ($n = 11$, 101.0 ± 5.7) and H3.3S31A ($n = 10$, 79.4 ± 8.0); Day 16–20: H3.3 ($n = 10$, 127.8 ± 10.2) and H3.3S31A

($n$ = 11, 91.9 ± 10.7). All data are Avg ± SEM. Unpaired $t$ test to compare the 2 individual datasets to each other. *: $P$ < 0.05, ns, not significant. Error bars represent the SEM. (**D**) Immunostaining of H3.3- and H3.3S31A-expressing testes with anti-Stat92E show decreased staining in H3.3S31A-expressing GSCs compared to H3.3-expressing GSCs. Asterisk: niche. Scale bars: 10 μm. (**E**) Quantification of Stat92E immunostaining signals in H3.3- and H3.3S31A-expressing testes. H3.3 ($n$ = 54 GSCs from 38 testes individual testes, 71.8 ± 6.6) and H3.3S31A ($n$ = 49 GSCs from 25 individual testes, 31.2 ± 3.7). Each data point is from an individual GSC. All data are Avg ± SEM. Unpaired $t$ test to compare the 2 individual datasets to each other. ****: $P$ < 0.0001. Error bars represent the SEM. The data underlying (**B**, **C**, **E**) can be found in S2 Table. GSC, germline stem cell; SEM, standard error of the mean; WT, wild-type.

differentially label old versus new histones in the context of the GSC cell cycle, as previously reported [29]. After a heat shock–induced genetic switch from eGFP- to mCherry-labeled histone expression, GSCs are allowed to undergo a complete cell cycle to incorporate new histones genome-wide, no matter whether the incorporation is replication dependent or replication independent (S1 Fig). We then studied eGFP (old) versus mCherry (new) histone segregation patterns during the second mitosis after the heat shock–induced switch.

Using live cell imaging, we first examined eGFP (old) versus mCherry (new) histone segregation patterns of the canonical histones H3 in mitotic GSCs (Fig 3A and S1 Movie). We quantified the ratios of the old and new histones between the 2 sets of sister chromatids in male GSCs at anaphase and telophase, using the 3D quantification method as shown previously [54,64]. Based on these quantifications, old H3 are significantly more enriched towards the set of sister chromatids that will be inherited by the future stem daughter cell (Fig 3E and 3F). New H3 are also asymmetric, to a lesser extent, between sister chromatids at this stage of mitosis (Fig 3F). By contrast, both old and new H3 display symmetric patterns in the mitotic SGs (Fig 3A' and S2 Movie). These live cell imaging results are consistent with the recent report that the overall nucleosome density is higher at the GSC-side sister chromatids than the GB-side sister chromatids in anaphase and telophase GSCs but is comparable between the 2 sets of sister chromatids in anaphase and telophase SGs [54].

As the incorporation of new histones is dynamically dependent on the cell cycle progression, we used different thresholds to classify the degree of asymmetry of old histones (Fig 3G). Based on this quantification, almost all mitotic GSCs (93%) showed an asymmetric pattern for old H3, and most mitotic GSCs (59%) also showed an asymmetric pattern for new H3 (Fig 3E–3G). These live cell imaging data in mitotic GSCs are largely consistent with previous studies using fixed sample imaging of postmitotic GSC–GB pairs [29,38]. The difference of new histone patterns between mitotic GSCs and postmitotic GSC–GB pairs is likely due to asynchronous initiation of the subsequent S phase between the GSC and daughter GB. The GB daughter nucleus enters DNA replication prior to the GSC daughter nucleus, resulting in increased new histone in the GB nucleus soon after exiting ACD [54].

Next, using similar dual-color labeling, we examined the segregation patterns of old versus new H3A31S and H3.3S31A during the second ACD of GSCs post-heat shock treatment. Interestingly, old versus new H3A31S showed less asymmetry between sister chromatids, distinct from WT H3 (Fig 3B and S3 Movie). Quantification of these live cell imaging results demonstrate that the 31st amino acid in the N-terminal tail is critical for proper segregation of histone H3 during ACD of GSCs (Fig 3F and 3G). We then studied the segregation pattern of WT histone variant H3.3, which does not exhibit such a globally asymmetric patterns (Fig 3C, 3F, and 3G and S4 Movie), consistent with previous report using fixed sample imaging in postmitotic GSC–GB pairs [29]. Furthermore, H3.3S31A also showed globally symmetric segregation pattern, indistinguishable from WT H3.3 (Fig 3D and S5 Movie). Quantification of the live cell imaging results indicate that old versus new H3A31S and H3.3S31A displayed globally much less asymmetric distribution patterns between sister chromatids in the mitotic GSCs (Fig 3E–3G). Taken together, these data demonstrate that the Ala31 residue in the N-terminal

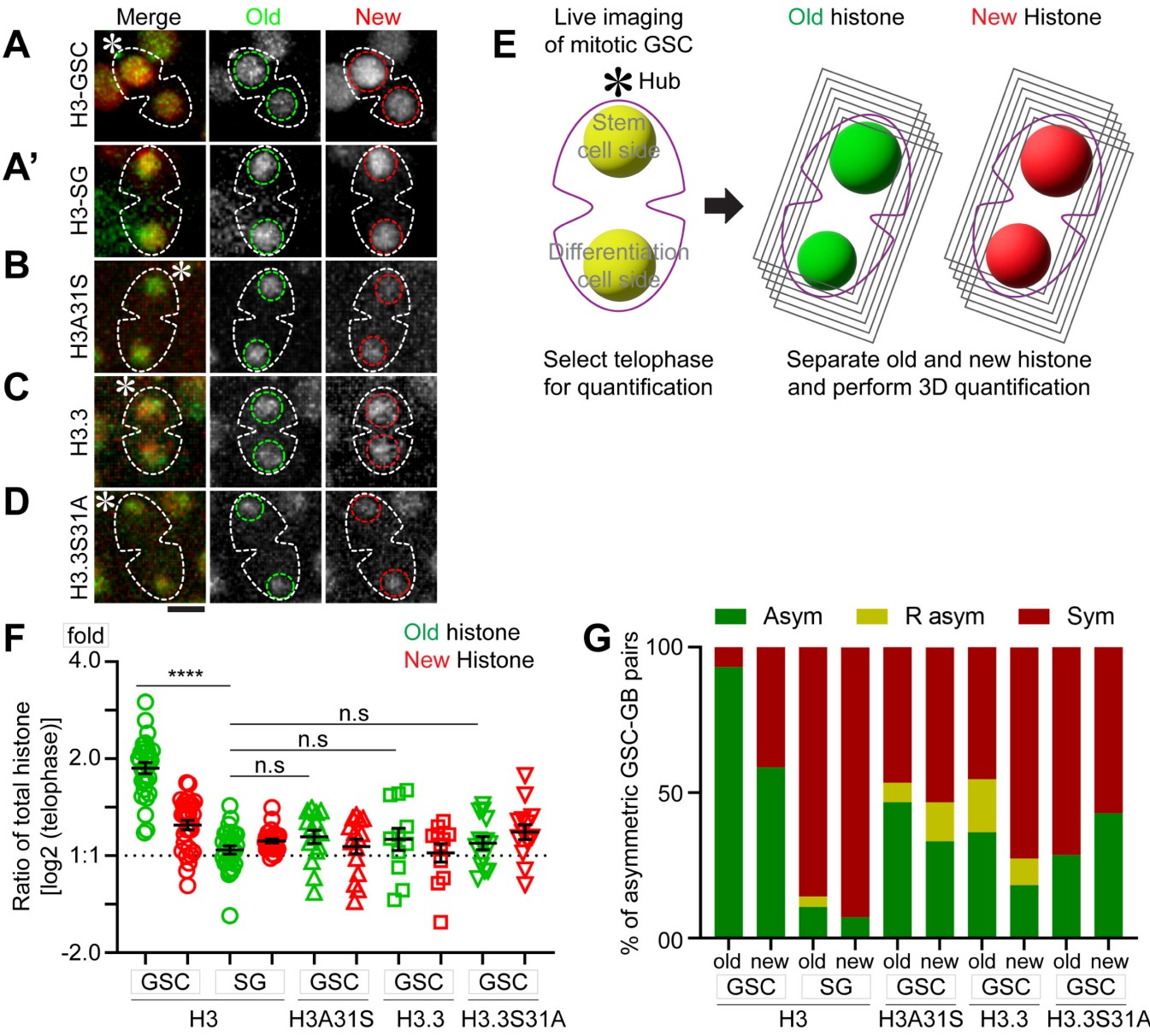

**Fig 3. Inheritance pattern of old versus new histone H3, H3A31S, H3.3, and H3.3S31S in mitotic GSCs using live cell imaging.** (**A**) Old H3 and new H3 inheritance in GSC. (**A'**) Old H3 and new H3 inheritance in SG. (**B**) Old and new mutant histone H3A31S inheritance in GSC. (**C**) Old and new histone variant H3.3 inheritance in GSC. (**D**) Old and new mutant histone H3.3S31A inheritance in GSC. All images are 3D reconstructed. Scale bar: 5 μm. (**E**) A schematic of 3D quantification methods of old histone (green) and new histone (red) in telophase male GSCs (see Materials and methods). (**F**) 3D quantification of old and new histone inheritance pattern by live cell imaging for GSCs and SGs in telophase (S3 Table). Old H3, GSC 1.91 ± 0.06 (*n* = 29), new H3, GSC 1.27 ± 0.04 (*n* = 29), old H3 SG 1.05 ± 0.03 (*n* = 28), and new H3 SG 1.11 ± 0.01 (*n* = 28); Old H3A31S, GSC 1.16 ± 0.05 (*n* = 15), new H3A31S, GSC 1.09 ± 0.05 (*n* = 15), Old H3.3, GSC 1.16 ± 0.09 (*n* = 11), new H3.3, GSC 1.04 ± 0.06 (*n* = 11), Old H3.3S31A, GSC 1.11 ± 0.05 (*n* = 14), new H3.3S31A, GSC 1.21 ± 0.07 (*n* = 14). ****$P < 10^{-4}$ by Mann–Whitney *t* test. The data underlying this panel can be found in S3 Table. (**G**) The fraction of asymmetric (Asym), symmetric (Sym), and inverse-asymmetric (R asym) histone inheritance pattern in H3, H3A31S, H3.3, and H3.3S31A expressing GSCs (see Materials and methods). GSC, germline stem cell; SG, spermatogonial cell.

tail of H3 is essential for proper H3 segregation during ACD, whereas the Ser31 residue in N-terminal tail of H3.3 does not alter the overall inheritance pattern of old and new histone variant H3.3 during ACD of GSCs.

### Old versus new H3A31S show more overlapping patterns than WT H3, while old versus new H3.3S31A show more separate patterns than WT H3.3 in prophase and prometaphase GSCs

Previously, we demonstrated that the asymmetric histone H3 distribution between sister chromatids are established during S-phase, followed by differential recognition by the mitotic machinery during M-phase of GSCs, in order to ensure their asymmetric inheritance patterns [30,65]. Furthermore, old H3-enriched sister chromatids condense to a higher degree than new H3-enriched sister chromatids in prophase to prometaphase GSCs, likely due to the higher nucleosome density and preferential PTMs on old histones, including phosphorylation and methylation on many residues [38,54,66,67].

To avoid a potential impact of asynchronous S-phase entry between GSC and GB nuclei on chromatin landscape and histone dynamics, which occur immediately after ACD, we closely examined old versus new histone patterns in prophase and prometaphase GSCs. We found that H3A31S showed more overlapping old versus new histone distribution patterns, compared with old versus new WT H3, which displayed more separate patterns (Figs 4A, 4B and S2). On the other hand, H3.3S31A-expressing GSCs displayed more separate old versus new histone domains compared with WT H3.3 (Figs 4C, 4D and S2).

Next, we quantified this separation versus overlapping patterns by measuring Spearman's rank correlation coefficient [68]. Here, a 0.0 correlation coefficient indicates no overlap, while the 1.0 correlation coefficient represents complete overlap between old and new histone

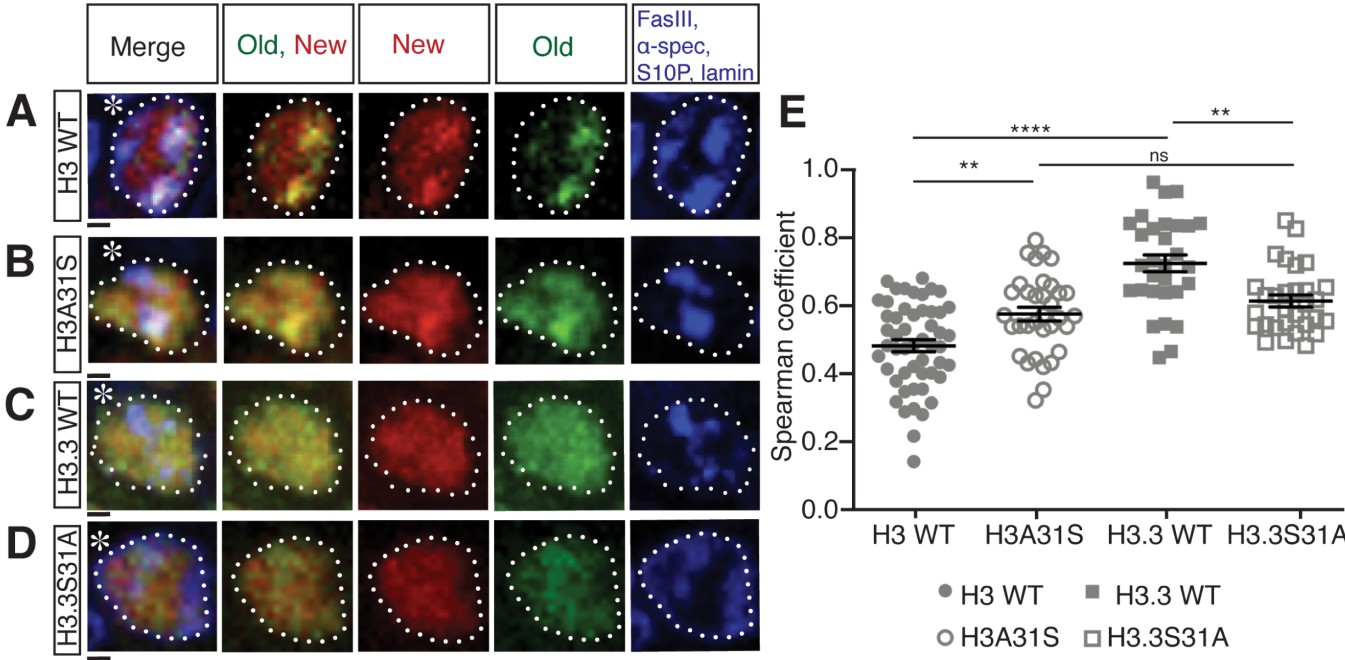

**Fig 4. Distribution patterns of old versus new histones in prophase or prometaphase GSCs.** (**A-D**) Old versus new WT H3 (**A**), H3A31S (**B**), WT H3.3 (**C**), H3.3S31A (**D**) distribution in late prophase and prometaphase GSCs labeled with H3S10P (blue, mitotic marker), showing old (green) and new (red) histone distribution patterns. Asterisk: niche. Scale bars: 1 μm. (**E**) Quantification of the Spearman's rank correlation coefficient showing different degrees of separation versus overlapping patterns between old and new histone in late prophase and prometaphase GSCs of H3 ($n = 52$, $0.47 \pm 0.02$), H3A31S ($n = 34$, $0.57 \pm 0.02$), H3.3 ($n = 30$, $0.72 \pm 0.03$), H3.3S31A ($n = 30$, $0.61 \pm 0.02$). All measurements are Avg ± SEM, which were taken from a single z-slice from the center of the nucleus, which were shown previously gave similar results to the measurements using all z-slices across the nucleus [68]. Individual data points are representative of each independent nucleus. Error bars represent the SEM. Pairwise ANOVA test with Bonferroni correction. ****: $P < 0.0001$, **: $P < 0.01$, ns, not significant. The data underlying this panel can be found in S4 Table. GSC, germline stem cell; SEM, standard error of the mean; WT, wild-type.

signals. Using similar quantification strategies, we examined prophase to prometaphase GSCs expressing WT H3, H3A31S, WT H3.3, and H3.3S31A, respectively (Fig 4E). First, a significant difference in the correlation coefficient was found between WT H3 and WT H3.3 (Fig 4A, 4C and 4E), as well as for WT H3 between asymmetrically dividing GSCs and symmetrically dividing SGs (S3 Fig). These results are consistent with their distinct inheritance patterns resulting from asymmetric GSC divisions (Fig 3A, 3C and 3E–3G), as well as with the previous results comparing the inheritance patterns between GSCs and SGs [29]. Second, old and new H3A31S showed a significantly higher correlation coefficient compared with WT H3, suggesting a more overlapping pattern between old and new H3A31S than that of old and new WT H3 (Fig 4A, 4B and 4E). Third, old and new H3.3S31A displayed a significantly lower correlation coefficient than that of old and new WT H3.3 (Fig 4C–4E). Interesting, changing the 31st Ala of H3 to Ser makes the distribution of old and new H3A31S more like WT H3.3, while switching the 31st Ser of H3.3 to Ala makes the distribution of old and new H3.3S31A more like WT H3 (Fig 4E). Collectively, these data suggest that the different 31st residues of the N-terminal tails between histone H3 and histone variant H3.3 play an important role in establishing the proper distribution of old versus new histones prior to M phase.

### Old histone-enriched H3K27me3 and H4K20me3 marks are more symmetrically distributed between replicative sister chromatids derived from H3A31S-expressing compared to WT H3-expressing early-stage male germ cells

Changes in old versus new histone distribution for both H3A31S and H3.3S31A mutants seem to be established prior to mitosis, as they are readily detectable in prophase to prometaphase GSCs (Fig 4). We next examined the old histone deposition or distribution pattern in earlier cell cycle phases, as old histone recycling during S phase or turnover during G2 phase likely contribute to old versus new histone patterns during M phase [20].

Based on the results that old versus new H3A31S display more overlapping pattern in prophase and prometaphase GSCs (Fig 4B and 4E) and globally symmetric segregation pattern in anaphase and telophase GSCs (Fig 3B, 3F and 3G), we hypothesize that old histone recycling in S-phase might be disrupted by expressing the H3A31S mutant histone. To understand how H3A31S may affect replication-dependent old histone recycling, we investigated the distribution of the tri-methylated lysine27 of histone H3 (H3K27me3) and the tri-methylated lysine20 of histone H4 (H4K20me3), which have been shown to be enriched on old histones [30,66,67,69]. Previously, we utilized an SRCF (superresolution imaging of chromatin fibers) method to visualize distribution of histone modifications between replicative sister chromatids. We found that H3K27me3 is enriched toward the leading strand on replicative sister chromatids, using chromatin fibers derived from WT H3-expressing early-stage germ cells (Fig 5A–5A') [30,70]. When we examined the distribution of H3K27me3 between replicative sister chromatids on chromatin fibers derived from H3A31S-expressing early-stage germ cells, we found that H3K27me3 displayed a more symmetric distribution (Fig 5B–5B'). By quantifying the chromatin fiber data, the H3K27me3 showed a significantly more symmetric pattern on chromatin fibers derived from H3A31S-expressing early-stage germ cells compared with fibers from WT H3-expressing cells (Fig 5E).

Additionally, we used another histone modification enriched on old histone H4 [67,69] to study its distribution on replicative chromatin fibers derived from early-stage germ cells expressing WT H3 (Fig 5C–5C') or H3A31S (Fig 5D–5D'). Consistent with the H3K27me3 data, H4K20me3 displayed significantly less asymmetric distribution on H3A31S-labeled fibers than H3-labeled ones (Fig 5F). Interestingly, the EdU pulse labeling on the replicative

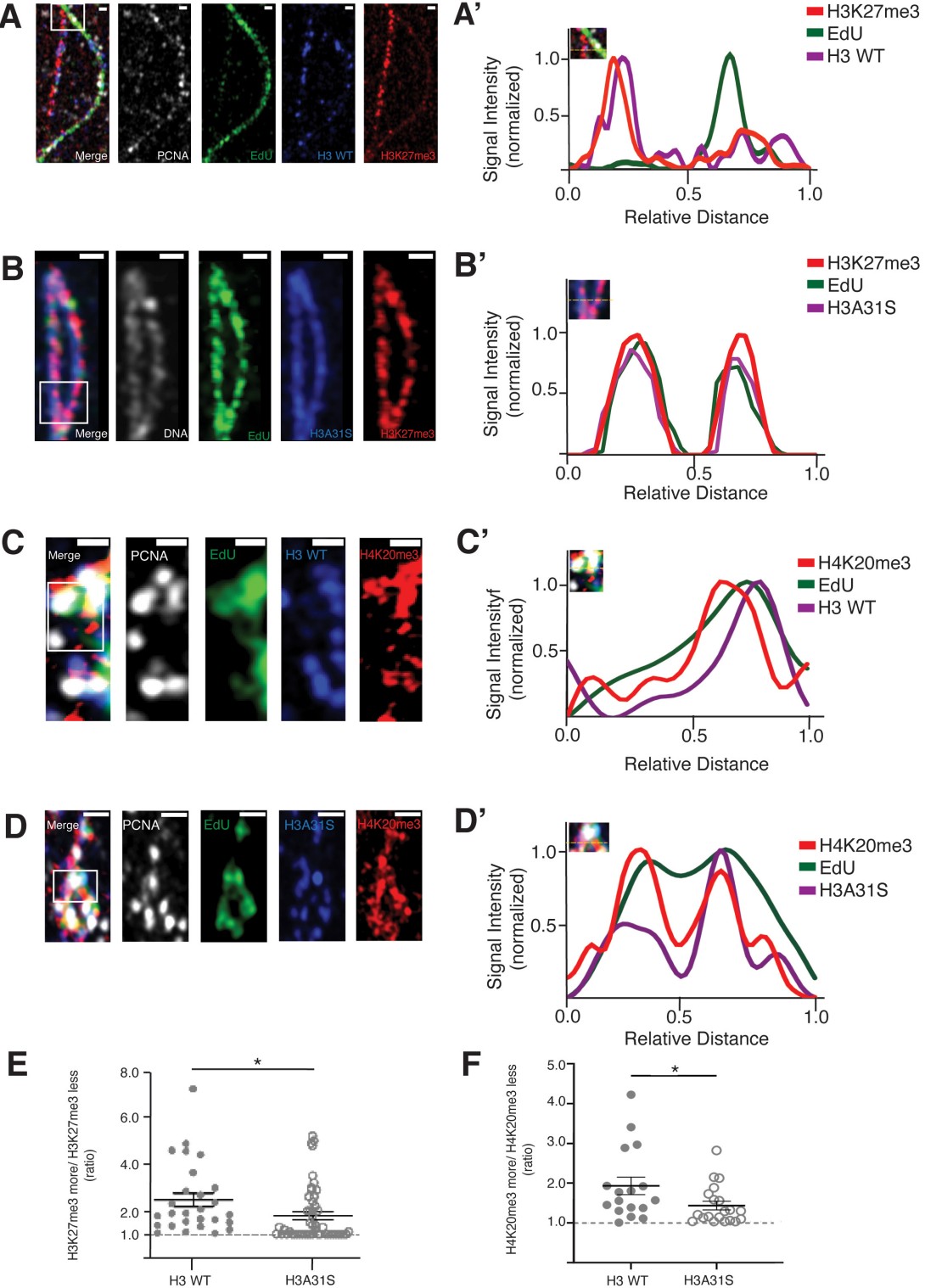

**Fig 5. More symmetric distribution of old histone-enriched H3K27me3 and H4K20me3 on replicative sister chromatids derived from *nos-Gal4>UAS-H3A31S* testes compared to *nos-Gal4>UAS-H3* testes.** (**A, B**) Airyscan images of chromatin fibers labeled with EdU showing distribution of H3K27me3 on replicating regions of (**A**) H3 WT or (**B**) H3A31S expressing testes. (**A', B'**) Line-plots show histone, H3K27me3, and EdU distribution across the replicating region (inset box with dashed yellow line). (**C, D**) Airyscan images of chromatin fibers labeled with EdU showing distribution of H4K20me3 on replicating

regions of (**C**) H3 WT or (**D**) H3A31S expressing testes. (**C'**, **D'**) Line-plots show histone, H3K27me3, and EdU distribution across the replicating region (inset box with dashed yellow line). Scale bars: (**A** and **D**) 1 μm, (**B**) 5 μm, and (**C**) 0.5 μm. (**E**) Quantification of the ratio of H3K27me3 fluorescence intensity on sister chromatid in H3-expressing testes ($n = 27$ replicating regions, $4.47 \pm 0.29$) and H3A31S-expressing testes ($n = 50$ replicating regions, $1.79 \pm 0.17$). Individual data points are representative of independent replicating regions. Error bars represent the SEM. Unpaired $t$ test to compare 2 individual datasets. *: $P < 0.05$. (**F**) Quantification of the ratio of H4K20me3 fluorescence intensity on sister chromatid in H3-expressing testes ($n = 17$ replicating regions, $1.933 \pm 0.2193$) and H3A31S-expressing testes ($n = 20$ replicating regions, $1.437 \pm 0.1087$). Individual data points are representative of independent replicating regions. Error bars represent the SEM. Unpaired $t$ test to compare 2 individual datasets. *: $P < 0.05$. The data underlying (**E**, **F**) can be found in S5 Table. SEM, standard error of the mean; WT, wild-type.

chromatin fibers derived from early-stage germ cells expressing WT H3 versus mutant H3A31S demonstrate less EdU asymmetry on H3A31S-labeled chromatin fibers than on WT H3-labeled chromatin fibers (Fig 5A and 5C versus Figs 5B, 5D and S4). The asymmetric EdU incorporation could reflect asynchronous synthesis between the leading strand and the lagging strand, which is likely compromised with H3A31S expression. Previously, we found that the lagging strand-enriched component PCNA and lagging strand-specific protein RPA-70, a highly conserved single-stranded DNA-binding protein, both have asymmetric distribution on early-stage germ cell–derived chromatin fibers [30]. In contrast, such asymmetries of EdU and PCNA were much less observed in symmetrically dividing cultured Kc cells [70]. In summary, these results suggest that the 31st residue at the N-terminal tail of H3 could be needed for old histone recycling during DNA replication.

## Rapid turnover of old H3.3S31A compared to WT H3.3 in G2-phase GSCs

On the other hand, because the histone variant H3.3 is known to be incorporated in a replication-independent manner [71], we hypothesize that the 31st residue at the N-terminal tail of H3.3 could be important for replication-independent histone turnover during G2 phase. To examine the turnover rate of H3.3 versus H3.3S31A, we measured the change in labeled old histone signals at 12 hours, 24 hours, and 36 hours, corresponding to the first, second, and third cell cycles following the heat shock–induced switch from eGFP- to mCherry-labeled histone expression in GSCs (Fig 6A) [29,30]. In order to pinpoint $G_2$ phase GSCs, given the very short $G_1$ phase in GSCs, we used anti-H3S10 phosphorylation (an M-phase marker) and EdU pulse labeling (an S-phase marker) to eliminate nuclei at M-phase or S-phase, respectively [54]. By measuring the eGFP fluorescent signals at each corresponding time points post-heat shock and normalizing them to the no-heat shock control, we found that the levels of old H3.3S31A decreased at a faster rate compared to WT H3.3 (Fig 6B and 6C), with almost all old H3.3S31A histone being turned over by 24 hours (at approximately 2 cell cycles) post-heat shock (Fig 6C and 6D–6D'). These results based on fixed samples are consistent with results using live cell imaging (S5 Fig). Together, these findings suggest the 31st amino acid plays a role in the turnover rate of H3.3, with this mutation accelerating replacement of old histone by new histone.

## ChIC-seq analyses of WT H3 and H3.3 versus mutant H3A31S and H3.3S31A in GSC-like cells

Next, we aimed to determine any salient genome-wide localization differences between WT histones H3 and H3.3 and their respective mutant counterparts. As each WT testis contains only 9 to 12 GSCs, we used a genetic manipulation to generate germline tumors enriched with GSC-like cells for genomic studies. Hyperactivation of the JAK–STAT pathway through

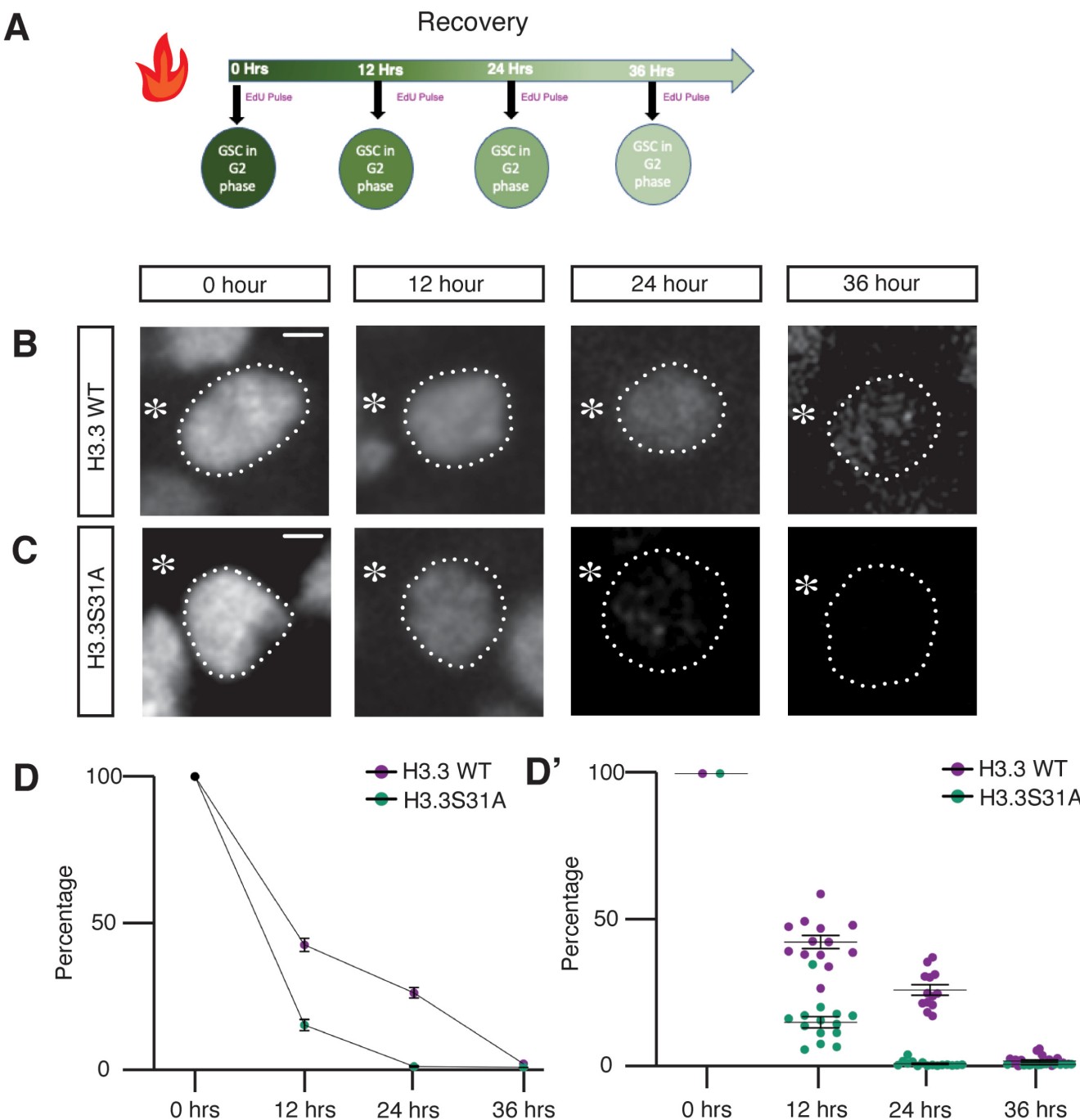

**Fig 6. Turnover of old H3.3 and H3.3S31A histones in G2 phase GSCs at different time points post-heat shock.** (A) Diagram of the time-course recovery experiment (12, 24, and 36 hours) post-heat shock to examine old histones in GSCs. A 30-minute EdU pulse incorporation was performed following the recovery period and right before tissue fixation. G2 phase GSCs were identified by elimination using M-phase (anti-H3S10ph) and S-phase (EdU) markers. (B, C) Confocal images of old histone distribution in (B) H3.3 and (C) H3.3S31A G2 phase GSCs. Asterisk: niche. Scale bars: 20 μm. (D) Quantification of old histone intensity in G2 phase GSCs at each time point post-heat shock recovery. Each data point is representative of average intensity at each time point for H3.3-expressing (12 hours: $n = 13$, 42.44 ±2.24; 24 hours: $n = 13$, 26.08 ± 1.77; 36 hours: $n = 18$, 1.79 ± 0.43) and H3.3S31A-expressing (12 hours: $n = 14$, 15.06 ± 1.93; 24 hours: $n = 15$, 0.86 ± 0.25; 36 hours: $n = 12$, 0.60 ± 0.04) GSCs, measurement from individual GSCs are shown in (D'). Error bars represent the SEM. The data underlying (D, D') can be found in S6 Table. GSC, germline stem cell; SEM, standard error of the mean.

overexpression of the Unpaired (Upd) ligand produces testes composed predominantly of GSC-like and cyst stem cell (CySC)-like cells [50,53,62]. We used this genetic background to obtain sufficient GSC-like cells to interrogate genomic histone patterns in a cell-specific manner using chromatin immunocleavage (ChIC) [72–74]. Here, we obtained germline specificity by coexpressing Upd with the GFP-tagged transgene of each 4 histone types using the *nos-Gal4* driver (S6 Fig). We compared the expression level of the transgenic histone to that of endogenous histones in *nos> Upd* testes. We found that the presence of transgenic histones did not affect the amount of endogenous histone (S7 Fig; [38]). We then applied the ChIC assay by targeting histone-GFP containing chromatin for digestion by MNase followed by preparation of soluble DNA for high-throughput sequencing (Fig 7A).

First, we confirmed that the transgenic histones incorporated throughout the genome (Fig 7B). Next, we were curious whether specific genomic regions were more likely to be differentially enriched for a histone or its mutant. We used the 9 genomic chromatin states previously defined by the ModENCODE consortium that included 42,126 DNA regions of variable sizes [75].We reasoned that because the chromatin states are associated with distinct combinations of chromatin modifications, that this would provide potential functional relevance to any differences we observe. To determine where a WT histone and its mutant counterpart differ in enrichment in these 9 chromatin states, we measured the number of reads located within a predetermined DNA region. A site is considered differentially enriched if the calculated $p$-value is smaller than 0.05 and the normalized number of reads for a histone type is >1.3-fold more than the other. This approach identified 378 chromatin state regions where H3 is more enriched than H3A31S, while 411 sites were identified as enriched for H3A31S compared with H3. In a similar manner, 148 sites have higher H3.3 localization, whereas H3.3S31A is more enriched at 164 sites.

Each of the identified differentially enriched genomic regions corresponded with one of 9 chromatin states previously defined by the ModENCODE project [75]. With this approach, we found that compared with WT H3, the H3S31A mutant is most frequently enriched at promoters and transcription start sites (TSSs, State 1, enriched with H3K4me3; Figs 7C and S8). Indeed, plotting an aggregate enrichment profile for these 2 histones centered at the TSS present in State 1 shows greater H3A31S localization compared with H3 (Fig 7E). On the other hand, there are more regions with a higher occupancy of H3.3S31A in transcriptionally silent intergenic regions, which is typically enriched for H3K27me3 (State 9; Fig 7D). Previous studies have reported that H3.3 is often associated with active transcription and enriched with PTMs such as H3K4me3 and H3K36me2 [76–80]. In contrast, PTMs associated with more repressive chromatin, such as H3K27me2/3 and H3K9me2/3, occur preferentially on H3 [77–79]. Here, our results demonstrate that swapping H3 to be more H3.3-like (H3A31S) causes this histone to behave more like H3.3, which is typically enriched in State 1 chromatin. The increase of H3.3S31A sites in State 9 domain is consistent with this histone behaving more like H3. Thus, swapping the single 31st residue in the histone tail may be sufficient to induce opposite changes in chromatin composition that underlie their opposing phenotypes in the male germline. To find more evidence at individual candidate genes, we measured H3.3 and H3.3S31A occupancy at the *stat92E* gene locus, which encodes a critical transcription factor to maintain GSCs as discussed above [44,50,53,61–63]. Interestingly, we found that there is a greater occupancy of H3.3S31A at the *stat92E* genomic region compared with H3.3 (Fig 7F). This replacement of H3.3 with an H3-like histone could change the accessibility of this gene locus to regulators, such as RNA polymerase II, for transcription initiation and/or elongation, which could explain the reduction in Stat92E protein abundance in the H3.3S31A-expressing GSCs compared to H3.3-expressing GSCs (Fig 2D and 2E).

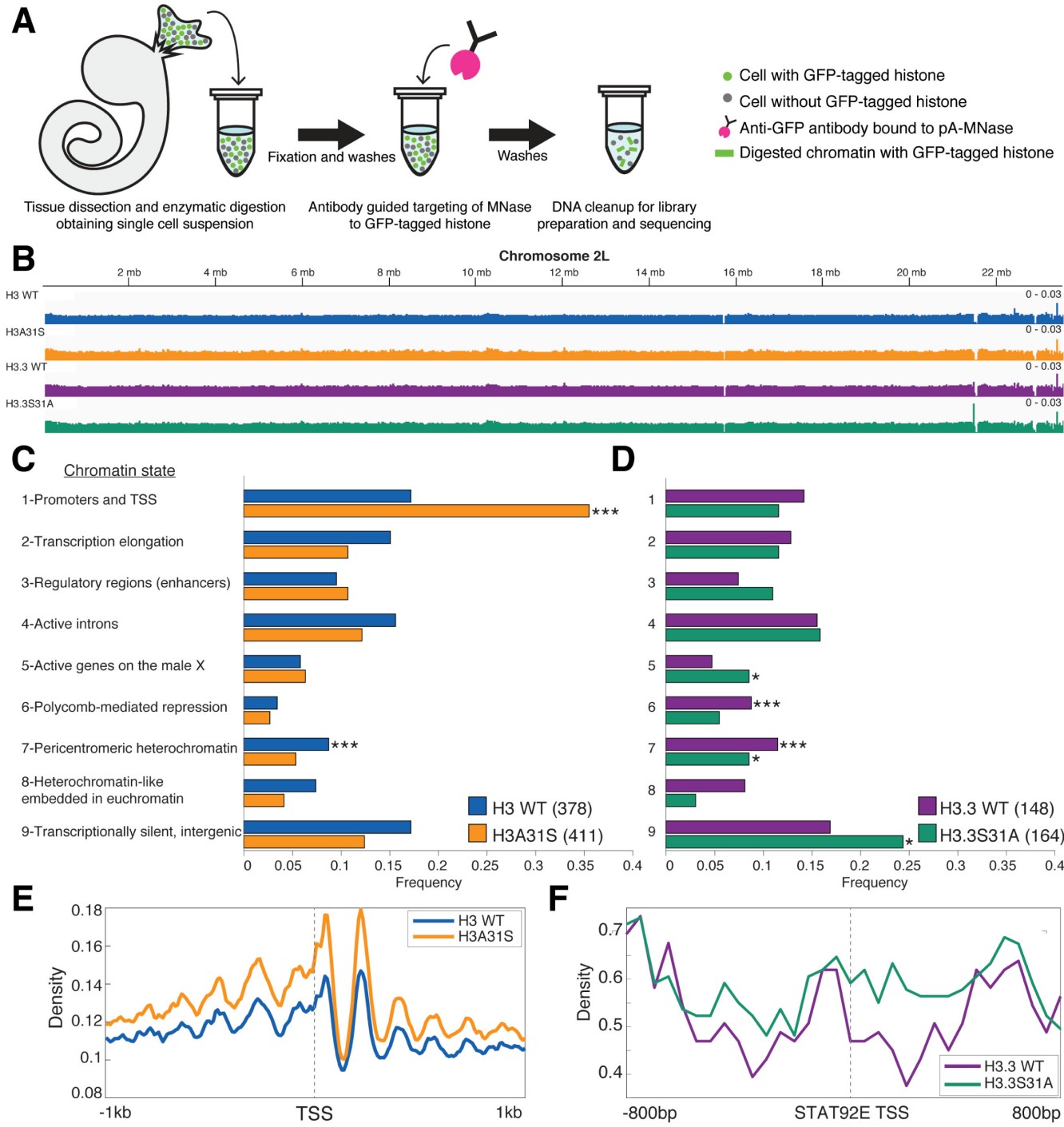

**Fig 7. The single residue substitution causes mislocalization of mutant histones to different chromatin contexts.** (**A**) Germline tumor testes with GSCs expressing GFP-tagged histones were used in a ChIC assay targeting MNase digestion to the tagged histones. The cleaved DNA was then prepared for next-generation DNA sequencing. (**B**) Genome browser view of Chromosome 2L showing incorporation of transgenic histones genome-wide. (**C, D**) Genomic enrichment was compared between each WT histone and the mutant counterpart within predefined regions of the specified chromatin states. The frequency of regions in each of the states that showed differential enrichment is indicated. *$P < 0.05$, ***$P < 0.001$. The data underlying these panels can be found in S7 Table. (**E**) Density of reads for H3 WT and H3A31S is plotted +/−1 kb centered on TSSs from chromatin state 1. (**E**) The density of H3.3 WT and H3.3S31A reads is shown +/− 800 bp centered at the *stat92E* TSS. ChIC, chromatin immunocleavage; GSC, germline stem cell; TSS, transcription start site; WT, wild-type.

## Discussion

In multicellular organisms, the endogenous *H3* genes are often found in gene clusters encoding canonical histones at multiple genomic locations [81], while the *H3.3* genes are typically singular ones in the genome [40,82]. Here, the two-color expression system uses the same transgene backbone, which eliminates the possibilities of their differential expression, pinpointing the key roles of the 31st amino acids located at the N-termini of H3 and H3.3 in an endogenous stem cell system. By swapping this single distinct amino acid in the H3's N-terminal tail to be H3.3-like, and H3.3's to be H3-like, we generated 2 hybrid proteins and uncovered their important roles in GSC maintenance and activity. In particular, the H3A31S (i.e., the H3.3 N-terminus + H3 C-terminus) disrupts biased old H3 recycling at the DNA replication fork (Fig 5), resulting in a more symmetric histone distribution pattern in the prophase and prometaphase GSCs (Fig 4A and 4B), as well as a more symmetric segregation pattern in the anaphase and telophase GSCs (Fig 3A and 3B). Interestingly, this mutant histone acts as an oncohistone, leading to accumulation of undifferentiated early-stage germ cells (Fig 1). On the other hand, the H3.3S31A (i.e., the H3 N-terminus + H3.3 C-terminus) results in a faster turnover rate compared to the WT H3.3 (Fig 6). Expression of this mutant histone leads to an opposite GSC loss phenotype, causing decreased male fertility over time (Fig 2). The GSC loss is likely attributed to declined Stat92E, a crucial stemness transcription factor. Interestingly, *stat92E* was recently shown to rely on asymmetric histone inheritance for its proper chromosomal architecture and gene expression in GSCs [83].

This tag switch method can differentially label any protein of interest in a spatiotemporally controlled manner. When applying to histones, the precise distinguishment between the 2 populations of histones (i.e., old versus new) is only applicable in the context of the actively ongoing cell cycle, and this precision declines over time. For example, the later expressed tagged histones are new during the first S phase but will become old during subsequent S phases. Contrastingly, the earlier expressed tagged histones represent old histones more precisely, but the turnover of these histones could cause this signal to diminish in following cell cycles. Additionally, as this tag switch occurs at the DNA level, it will take time for the switch to be reflected at the protein level, considering RNA stability and protein perdurance of old histone, as well as the time needed for the new histone gene to be transcribed, translated, and properly localized. Thus, this method is more appropriate for cell types with relatively long cell cycles such as *Drosophila* male GSCs, and it is important to monitor the production and incorporation of new histone in a time-course experiment to understand the dynamics of the tag switch at the protein level in the context of ongoing cell cycles.

H3 and H3.3 are 2 of the most conserved proteins among all eukaryotic organisms [82]. Intriguingly, unicellular organisms such as yeast only have H3.3-like histones, whereas multicellular organisms have both H3.3-like and H3-like histones [40]. Notably, the primary sequences of H3 and H3.3 differ at the 31st position at the N-terminal tails and the 87th to 90th amino acids at their C-terminal core regions [84]. It has been shown that H3 and H3.3 exhibit distinct interactions with histone chaperones due to their differences at the C-termini, which have been proposed to be responsible for their different genomic distributions [32,42,85–88]. On the other hand, the N-terminal tails of histones have the potential to undergo extensive PTMs that alter local chromatin landscape and association with other chromatin regulators [17]. To date, the distinct roles of the 31st amino acid in H3 and H3.3 remain largely unclear. Surprisingly, based on our results, the 31st amino acid is also responsible for the distinct behaviors between H3 and H3.3, including their segregation patterns during ACD of GSCs (Fig 3), their distribution patterns as old versus new histones (Fig 4), and their differential genome occupancy (Fig 7). One hypothesis for the function of the 31st amino acid could

be its role in the interactions between H3/H3.3 and H4, thus playing a role in nucleosome dynamics. This hypothesis is consistent with the observation that the H3.3S31A mutation leads to faster old histone turnover at the G2 phase (Fig 6). For H3, the 31st amino acid Ala may play a role in interacting with replication components and/or chaperones during S-phase, ensuring faithful recycling of old histones on sister chromatids during replication, consistent with the finding that the H3A31S mutation changes the asymmetric old histone distribution at replication loci (Fig 5). An alternative but not mutually exclusive hypothesis is that the 31st position may carry a PTM at the Serine residue in H3.3, which has the potential to be modified by a phosphoryl group [89]. The phosphorylation may hold the key to how the 31st residue of H3.3 acts to maintain germline identity and normal activity, without which GSCs fail to maintain themselves, resulting in deteriorated fertility over time (Fig 2). Future biochemistry experiments will provide insight into the potential enzymes that modify H3.3S31 and the putative effector that acts through H3.3S31.

A key developmental biology question is how epigenetic mechanisms direct specification of distinct cell fates in the daughter cell derived from ACD. Incorrect transmission of epigenetic information can result in diseases such as cancer, tissue degeneration, or infertility. Recently, somatic mutations affecting H3 and H3.3, termed oncohistones, have been characterized in aggressive cancers [36,37], further emphasizing the importance to understanding the specific roles of H3 and H3.3 in a developmental context [90]. We observed early germ cell tumors in the H3A31S mutant and GSC loss in H3.3S31A mutant. The *Drosophila* germline, and its many genetic tools and resources, provides a model to study this mechanistic link between epigenetic inheritance and tumor initiation and germ cell integrity. Furthermore, our novel finding that the H3A31S is an oncohistone in the male germline, provides a model to study other such histone mutations, in order to gain insight on oncohistones to better understand various human diseases including fertility and early-stage developmental defects.

## Materials and methods

### Fly strains and husbandry

Fly stocks were raised using standard Bloomington medium at 18˚C, 25˚C, or 29˚C as noted. The following fly stocks were used: *hs-flp* on the X chromosome (Bloominton Stock Center BL-26902), *nos-Gal4* on the second or the third chromosome [43], *UASp-FRT-H3-GFP-PolyA-FRT-H3-mCherry* on the second chromosome, *UASp-FRT-H3.3-GFP-PolyA-FRT-H3.3-mCherry* on the third chromosome, *UASp-FRT-H3A31S-GFP-PolyA-FRT-H3A31S-mCherry* on the third chromosome, *UASp-FRT-H3.3S31A-GFP-PolyA-FRT-H3.3S31A-mCherry* on the second chromosome.

For fly lines used in the ChIC experiment, standard fly genetics was used to introduce the switchable dual-tagged histone cassettes into a genetic background containing a *UAS-upd* transgene [91], which was kindly provided by Dr. Stephen Dinardo (University of Pennsylvania, USA). This produced the following 4 lines:

w[1118]; UASp-FRT-H3-EGFP-PolyA-FRT-H3-mCherry, UAS-upd /CyO

w[1118]; UASp-FRT-H3.3-EGFP-PolyA-FRT-H3.3-mCherry, UAS-upd /CyO

w[1118]; UASp-FRT-H3.3S31A-EGFP-PolyA-FRT-H3.3S31A-mCherry, UAS-upd /CyO

*w[1118]; UAS-Upd /CyO; UASp-FRT-H3A31S-EGFP-PolyA-FRT-H3A31S-mCherry/TM6B.*

Males from the first 3 lines were individually crossed to the *hs-flp; nos>Gal4* females, which were reared at 25˚C in standard molasses bottles. After eclosion, male progenies with the following genotypes were allowed to age for approximately 2 days before dissection and sample preparation for ChIC-seq:

hsFLP; UASp-FRT-H3-EGFP-PolyA-FRT-H3-mCherry, UAS-Upd / nos>Gal4

hsFLP; UASp-FRT-H3.3-EGFP-PolyA-FRT-H3.3-mCherry, UAS-Upd / nos>Gal4

hsFLP; UASp-FRT-H3.3S31A-EGFP-PolyA-FRT-H3.3S31A-mCherry, UAS-Upd / nos>Gal4

The last line was crossed to hsflp; *nos(green eye)-Gal4/CyO; +/MKRS* [68,92,93]. The *nos (green eye)-Gal4* line was generously provided by Dr. Daniela Drummond-Barbosa (Johns Hopkins Bloomberg School of Public Health and currently University of Wisconsin-Madison, USA). The following genotyped males were used:

hsFLP; UAS-Upd / nos>Gal4; UASp-FRT-H3A31S-EGFP-PolyA-FRT-H3A31S-mCherry/+.

## Generation of fly strains with different switchable dual-color transgenes

Standard procedures were used for all molecular cloning experiments. Enzymes used for plasmid construction were obtained from New England Biolabs (Beverly, MA). H3A31S and H3.3S31A point mutations were generated with quick change site-directed mutagenesis kit (Agilent Technologies 200521) according to manufacturer's instructions, based on the pBluescript plasmids containing WT H3 and H3.3 sequences described in [29]. The new histone sequences, including H3-mCherry, H3.3-mCherry, H3A31S-mCherry, and H3.3S31A-mCherry, were recovered as an XbaI flanked fragment from pBluescript-new histone plasmids and were subsequently inserted into the XbaI site of the UASp plasmid to construct the UASp-new histone plasmids. The old histone sequences, including H3-EGFP, H3.3-EGFP, H3A31S-EGFP, and H3.3S31A-EGFP, were inserted to pBluescript-FRT-NheI-SV40 Poly-A-FRT plasmid at the unique NheI site. The entire NotI-FRT-old histone-EGFP-SV40 Poly-A-FRT-EcoRI sequences were then subcloned into the UASp-new histone-mCherry plasmid digested by NotI and EcoRI. The final UASp-FRT-old histone-EGFP-PolyA-FRT-new histone-mCherry plasmids were introduced to w1118 flies by P-element-mediated germline transformation (Bestgene). The *UASp-FRT-H3-EGFP-PolyA-FRT-H3-mCherry* transgenic fly lines were used in [30,65]. Transgenic flies with the following transgenes were newly generated in studies reported here: *UASp-FRT-H3.3-EGFP-PolyA-FRT-H3.3-mCherry, UASp-FRT-H3A31S-EGFP-PolyA-FRT-H3A31S-mCherry* and *UASp-FRT-H3.3S31A-EGFP-PolyA-FRT-H3.3S31A-mCherry*.

## Heat shock scheme

Male flies with UASp-dual color transgene were paired with females containing the *nos-Gal4* driver [43]. Flies were raised at 18°C or 25°C throughout development until adulthood, as noted. For adult males: Prior to heat shock in a circulating 37°C water bath for 90 minutes, 0 to 3 day old were transferred to vials that had been air dried for 24 hours. Vials were submerged under water up to the vial plug in the water bath. Following heat shock, vials were recovered in a 29°C incubator for indicated time before dissection for immunostaining experiments.

## Protein extraction and immunoblotting

For the immunoblot, a parental cross between female flies with the *nos-Gal4* driver and males containing the UASp-dual color histone transgenes was used to obtain male progenies with the appropriate germline-expressing histone transgene. Testes were dissected in 1XPBS from 5 (Upd overexpressing tumor) males before transferring to 10 μL of 1XRIPA buffer (Boston Bioproducts, #BP-115). After briefly boiling for 5 minutes at 100°C and homogenizing, the concentration of protein lysates was then measured (Thermo Fisher Scientific Qubit Protein Assay #Q33211). An equal volume of Novex Tricine SDS sample buffer 2X (Thermo Fisher Scientific #LC 1676) was added prior to a second boiling step at 100°C. A total of 5 tumor

pairs per genotype were loaded onto a 10% to 20% Tricine gel (Thermo Fisher Scientific #EC6625BOX). Proteins were separated in Tricine SDS sample buffer for 85 minutes at constant 120V (Thermo Fisher Scientific #LC1676.) The proteins were next transferred onto methanol-activated PVDF (Thermo Fisher Scientific #LC2005) membrane in transfer buffer (Boston BioProducts #BP-190) in the cold room at a constant 30 V for 1.5 hours. After successful transfer of proteins, the membrane was blocked for 30 minutes at room temperature in 5% BSA, TBST (Cell Signaling #9998S, Boston BioProducts #IBB-180). Blots were incubated overnight at 4°C with gentle rotation in primary antibodies: Rabbit anti-H3 (Abcam #ab1791, 1:1,000), Rabbit anti-H3.3 (Abcam #ab176840, 1:1,000), or Chicken anti-GFP (Abcam #ab13970, 1:1,000) in 5% BSA, 1XTBST. The anti-H3 immunoblot was stripped following the manufacturer's instructions with Restore Western Blot stripping buffer (Thermo Fisher Scientific #21059) reagent and reprobed for GFP. The immunoblots were next washed 3X5 minutes in 1XTBST at room temperature. Secondary antibody incubation was performed for 2 hours at room temperature in 1% BSA, 1XTBST (HRP-conjugated Mouse anti-Rabbit, Cell Signaling #5127, 1:2,000; HRP-conjugated Goat anti-Chicken, Abcam #97135, 1:1,000). Following secondary incubation, the blots were washed once again 3X5minutes in 1XTBST at room temperature. The immunoblots were developed following the addition of ECL substrate (Abcam #ab133406) and chemiluminescence visualized with a Syngene G:Box.

### Live cell imaging

Live cell imaging was performed between 12 and 30 hours after heat shock treatment, as detailed in [94]. To examine the inheritance pattern of histones during asymmetric GSC divisions, we conducted live cell imaging with high temporal resolution (e.g., 5-minute interval as mention in the S1–S5 Movie legends). To perform live cell imaging, adult *Drosophila* testes were dissected in a "live cell medium" as reported previously [65]. Live cell medium contains Schneider's insect medium with 200 μg/ml insulin, 15% (vol/vol) fetal bovine serum (FBS), 0.6× pen/strep, with pH value at approximately 7.0. Testes were then placed on a Poly-D-lysine coated FluoroDish (World Precision Instrument), which contains the live cell medium as described. All movies were taken using spinning disc confocal microscope (Zeiss) equipped with an evolve camera (Photometrics), using a 63× Zeiss objective (1.4 NA) at 29°C. The ZEN 2 software (Zeiss) was used for acquisition with 2 × 2 binning. All videos for live cells are shown in S1–S5 Movies. WT histone H3 and mutant H3A31S movies were acquired between 22 and 30 hours post heat shock treatment. WT H3.3 and mutant H3.3S31A movies were acquired between 12 and 25 hours post heat shock treatment.

### Define conventional and inverted asymmetries, and symmetry

For histone inheritance patterns in telophase, we have used 8-cell SG as a control. To define different categories of symmetric, conventional asymmetric and inverted asymmetric for histone inheritance in telophase GSCs, we used those ratios in symmetrically dividing SGs (SG1/SG2) to define the symmetric range. For example, in telophase, ratios above {SG1/SG2 mean + STD [1.053 + 0.155 = 1.208 (∼1.20)]} are called "conventional asymmetry," ratios below {SG2/SG1 mean − STD [0.949–0.155 = 0.794 (∼0.8)]} are called "inverted asymmetry," and ratio between 0.8 and 1.2 are called "symmetry." Related to Fig 3.

### Immunostaining (whole mount)

Immunofluorescence staining was performed as described previously [29,30,38,65]. For whole mount immunofluorescence staining in Figs 1, 2, 4 and 6, testes from 0- to 2-day-old flies were dissected in Schneider's *Drosophila* medium (Gibco, catalog # 21720001). Samples were then

fixed in 4% formaldehyde in phosphate-buffered saline (PBS) with 0.1% Triton X-100 for 10 minutes at room temperature. Samples were washed thrice for 10 minutes per wash in PBST. Samples were incubated for 24 hours at 4°C with primary antibodies in PBST with 3% bovine serum albumin. Primary antibodies used were anti-Fasciclin III (Fas III) (1:200, DSHB, AB_528238), anti-GFP (1:1,000; Abcam ab 13970), anti-H3K27me3 (1:200; Millipore 07–449), anti-H4K20me3 (1:200, Thermo Fisher, Cat #701777), anti-α Spectrin (1:200, DSHB, AB_528473), anti-Vasa (1:100, Santa Cruz, SC-30210), anti-Armadillo (1:200, DSHB, AB_528089), anti-Lamin (1:200, DSHB, AB_528336), anti-PCNA (1:200; Santa Cruz sc-56), anti-mCherry (1:1,000; Invitrogen M11217), anti-Stat92E (1:200, gift from Denise Montell, University of Santa Barbara, CA, USA), anti-H3S10ph (1:1,000, Abcam, AB 14995). Following primary antibody incubation, the sample was washed 3 times for 10 minutes in PBST and then incubated in 1:1,000 dilution of Alexa Fluor–conjugated secondary antibody (from Molecular Probes) overnight. Samples were washed 3 times for 10 minutes in PBST and mounted for microscopy in Vectashield antifade mounting medium (Vector Laboratories, Cat#H-1400) with/without DAPI. Slides (Fisherbrand Superfrost Plus Microscope Slides) and covers were examined using the Leica DMi8 confocal microscope, Zeiss LSM 700 confocal microscope, Zeiss LSM 800 confocal microscope with Airyscan with 63× oil immersion objective. Images from individual testis were analyzed using Fiji software. A circle was drawn around the region of interest and fluorescent signal was measured using the Fiji software. The total amount of the fluorescence signal in the nuclei was calculated by summing the individual Z-stacks in the nuclei. Figures were prepared for publication using Adobe Illustrator.

## Immunostaining (squash method)

Immunostaining of unpaired overexpression tumor testes was done using a squash method for S6 Fig. First, approximately 5 testis pairs from 0 to 3 day of male flies were dissected in warm Schneider's *Drosophila* medium. The testes were transferred to 10 μL of 1XPBS on a microscope slide. Each tumor was gently punctured to allow cells to spill from the tissue. The samples were then covered with a coverslip. The slides were next submerged in liquid nitrogen for at least 2 minutes. After removing the coverslip with a razor blade, the slides were immediately placed in prechilled 95% ethanol and incubated at −20°C for 10 minutes. Next, approximately 50 μL of fixative (1% formaldehyde in 1XPBST) was placed over the tissue and allowed to sit for 3 minutes at room temperature. Following fixation, the slides were rinsed quickly in a Coplin jar 3 times with 1XPBST.

To permeabilize the tissue, the slides were washed twice for 15 minutes each in 1XPBST containing 0.5% sodium deoxycholate. Next, the slides were rinsed in 1XPBST for 10 minutes. At this point, the slides were ready for primary antibody incubation. About 20 μL of primary antibody diluted in 3%BSA/PBST was prepared for each slide. The antibodies used in this preparation are as follows: chicken anti-eGFP (1:1,000, Abcam #13970) and guinea pig anti-Traffic Jam (1:1,000, gift from Dr. Mark Van Doren's laboratory). Primary antibody was added to each slide and covered with parafilm before placing in a humid chamber overnight at 4°C.

After primary antibody incubation, the slides were quickly rinsed twice in 1XPBST prior to being washed 3 times for 5 minutes on a rotator. Secondary antibodies were diluted (1:1,000) in 5% NGS in 1XPBST and incubated in darkness overnight in a humid chamber at 4°C. Finally, the slides were rinsed twice in 1XPBST followed by three 5-minute washes. The slides were mounted in Vectasheild antifade mounting media containing DAPI prior to imaging.

## Incorporation of EdU

EdU labeling was incorporated using the Click-iT Plus EdU Alexa Fluor 647 Imaging Kit (Invitrogen C10640) or Click-iT Plus EdU Alexa Fluor 594 Imaging Kit (Invitrogen C10339)

according to manufacturer's instructions. Dissected testes were incubated in in Schneider's *Drosophila* medium (Gibco, catalog # 21720001) with 10 μM EdU for 30 minutes at room temperature. The testes were later fixed and incubated with primary antibody, as described in the immunostaining protocol above. Fluorophore conjugation to EdU was performed according to the manufacturer's instructions, prior to secondary antibody incubation.

## Chromatin fiber preparation with nucleoside analog incorporation and immunostaining

Testes were dissected in Schneider's *Drosophila* medium at room temperature and incubated in Schneider's medium containing 10 μM EdU analog (Invitrogen Click-iT EdU Imaging Kit, catalog # C10640, #C10339). Testes were incubated for 10 minutes, rotating at room temperature. Following incubation with Edu analog, testes were incubated in the dissociation buffer (Dulbecco's PBS with $Mg^{2+}$ and $Ca^{2+}$ with collagenase/dispase (MilliporeSigma) added to a final concentration of 2 mg/ml) in a 37˚C water bath for 10 minutes. Cells were pelleted at 1,000*g* for 5 minutes, after which the dissociation buffer was drained using a cell strainer. Cells were suspended in 40 μl of lysis buffer (100 mM NaCl, 25 mM Tris-base, 0.2% Joy detergent (pH 10)). After resuspension, 20 μl of lysis buffer/cell mixture was transferred to a clean glass slide (Fisherbrand Superfrost Plus Microscope Slides) and allowed to sit in lysis buffer until cells were fully lysed (approximately 5 minutes). Around 10 μl of sucrose/formalin solution (1 M sucrose; 10% formaldehyde) was then added and incubated for 2 minutes. A coverslip (Fisherbrand Microscope Cover Glass, 12-545-J 24 × 60 mm) was placed on top of the lysed chromatin solution, and then the slide was transferred immediately to liquid nitrogen and allowed to sit for 2 minutes. The cover slip was then removed with a razor blade, and the slide was transferred to cold (−20˚C) 95% ethanol for 10 minutes. Next, the slide was incubated with fixative solution 0.5% formaldehyde in 1× PBST for 1 minute. The fixative solution was drained, and the slides were placed into a Coplin jar containing 50 ml 1× PBS. Slides were washed twice with 50 ml 1× PBS each time and placed in a humid chamber with 1 ml of blocking solution (2.5% BSA in 1× PBST) for 30 minutes of preblocking. Blocking buffer was then drained, and primary antibodies were added for incubation overnight at 4˚C. Slides were then washed twice with 50 ml 1× PBS and incubated with secondary antibodies for 2 hours at room temperature. Slides were then washed twice with 50 ml 1× PBS and mounted with Vecta shield antifade mounting medium (Vector Laboratories, Cat#H-1400) with DAPI.

## Fecundity test

Mutation expressing flies and WT flies were crossed to 3 yellow white female virgin female flies. Every 5 days, the male was crossed to 3 new, yellow white female virgin female flies. The progenies were counted on the first 3 days after eclosure.

## Image analysis methods

**Histone turnover in GSCs.** For histone turnover experiments, flies were heat-shocked as described above, and allowed to recover for 12, 24, or 36 hours, followed by incubation with EdU analog for 30 minutes and fixed as described. The eGFP fluorescent signals representing old histone in the G2-phase GSCs were quantified as described above and normalized to the no-heat shock control as a percentage of fluorescence, at each corresponding time point (12, 24, and 36 hours) post-heat shock, respectively. All ratios are reported as average ± standard error of the mean (SEM).

**Quantification of histone or posttranslational modifications on replicating sister chromatids.** Quantification for fibers were done using the Fiji software. To measure protein

values across sister chromatids, replicated regions were subdivided into 2 μM sections along the length of the fiber and were measured for fluorescence intensity as reported previously [30,70]. We then divided fluorescence intensity from the brighter sister chromatid fiber segment by the fluorescence intensity from the less bright sister chromatid fiber segment, to generate a ratio of the relative difference between sister chromatids. At the replication junction, line plots were drawn across sister chromatids to measure average fluorescence intensities at a specific region.

**Quantification of old versus new histone overlap.** Quantification was conducted as described in [68]. A single Z-slice containing the most chromatin information was utilized for quantification. A region of interest was drawn around the cell of interest using the circle tool. The FIJI Coloc2 plugin to analyze overlap between red and green channels with settings PSF (3) and Costes iterations (10). Values for Spearman correlation for each histone are plotted in graph.

## Chromatin immunocleavage (ChIC) sequencing

**Testis dissection and generation of single cell solution.** Flies of the correct genotype were briefly rinsed in 75% ethanol, before proceeding to dissection. Testis tumors were removed in freshly prepared, filter sterilized Schneider's media (Gibco #21720–024) supplemented with 10% FBS (Thermo Fisher #16140071). Only the largest testes, with minimal differentiating germline, were used in the preparation. For each genotype, an equivalent number of testis pairs was used in each of 3 biological replicates ranging from 18 to 20 pairs. After dissection, testes were moved to a microcentrifuge tube containing 200 to 300 μL of tissue digestion buffer (TrypLE Express (Gibco #12605–020) supplemented with 2 mg/mL collagenase (Sigma #C9407). Tissue was digested to completion in a 37˚C water bath for 10 to 15 minutes, with agitation every 2 to 3 minutes.

Following tissue digestion, suspended cells were subject to sequential filtering into a new tube first with a 40-μm nylon mesh cell strainer (CellTreat #229482), followed by a 10-μm cell strainer (pluriSelect #43-10010-40). Cells were pelleted by centrifugation at 1,200 rpm at 10-minute intervals. Digestion buffer was removed carefully, so as not to disturb the pelleted cells. Centrifugation and buffer removal were repeated until all buffer was removed. The pelleted cells were resuspended in 400 μL Schneider's media containing 10% FBS. Fixation began with the addition of 16% formaldehyde (Thermo Scientific #38906) added at 1/15th the volume to obtain a final concentration of 1%. The tubes were incubated at room temperature for 5 minutes with end-over-end rotation. The fixation reaction was quenched with the addition of 1/10 the volume of 1.25 M glycine. Again, cells were rotated end-over-end for 5 minutes at room temperature. The fixation solution was next removed by pelleting cells at 1,320 rpm for 7 minutes at 4˚C. From this point on, the cells were kept on ice. The fix solution was removed, and 500 μL cold 1XPBS was added to the pellet. Centrifugation was repeated, and 1XPBS added once again. This wash step was repeated for a third time. After the last wash, cells were resuspended in 20 μL 1XPBS. To obtain cell count using trypan Blue (Corning #25-900-CI), 1 μL of cells was removed. While counting, the cells remained on ice.

**Preparation of antibody-ProteinA-Mnase complex, chromatin release, and incubation.** Chromatin digestion of GSC-specific H3-GFP chromatin was done by antibody-guided MNase digestion. For a single replicate, the ProteinA-MNase and antibody complex was prepared by combining 4 μL antibody binding buffer (1X Tris EDTA (pH 7.5), 150 mM NaCl, 0.1% Triton-X 100), 1 μL anti-GFP antibody (Abcam ab290), and 3 μL PA-MNase enzyme. The mixture was allowed to incubate on ice for 30 minutes. Meanwhile, after noting the number of cells, the entire cell pellet was resuspended in 500 μL RIPA (1X Tris, EDTA (pH 7.5),

150 mM NaCl, 0.2% SDS, 0.1% Sodium Deoxycholate, 1% Triton-X 100) and incubated at room temperature for 10 minutes. Fixed cells were then pelleted by spinning at room temperature for 5 minutes at 3,000 rpm. The RIPA buffer was removed, and the cells washed once with 500 µL antibody binding buffer. Cells were pelleted once again prior to being resuspended in 100 µL antibody binding buffer. The 100 µL of cells in antibody binding buffer were added to the tubes containing the 8 µL of Ab+PA-MNase complex. The cells were mixed gently by pipetting before incubating on ice for 60 minutes.

**Washes to remove excess antibody complex.** Following incubation, the cells were subject to a series of washes to remove excess antibody complex. All centrifugation steps were performed at 3,000 rpm for 5 minutes at 4˚C. First, cells were spun down to remove antibody binding buffer. They were then washed with 500 µL high salt wash (1X Tris, EDTA (pH 7.5), 400 mM NaCl, 1% Triton-X 100), incubating for 1 minute at room temperature before spinning down. The high salt wash was repeated for a total of 3 washes. After the third wash, cells were washed in 200 µL of rinsing buffer (10 mM Tris (pH 7.5), 10 mM NaCl, 0.1% Triton-X 100) followed immediately with spinning down.

**Perform MNase digestion.** After removing the rinsing buffer, the cell pellet was resuspended in 40 µL of RSB (20 mM Tris (pH 7.5), 10 mM NaCl, 2 mM $CaCl_2$, 0.1% Triton-X 100). The cells were then incubated at 37˚C in a water bath for precisely 3 minutes. Next, 80 µL of stop buffer (20 mM Tris (pH 8), 10 mM EGTA, 20 mM NaCl, 0.2% SDS) was added to the reaction, followed by 1 µL of 19 mg/ml Proteinase K (NEB #P8107S). After mixing well, the solution was incubated overnight at 65˚C.

**Purification of cleaved DNA.** The cleaved DNA was purified from the cell lysate following manufacturer's recommendation using the MinElute Reaction Cleanup kit (Qiagen #28204). The DNA was eluted in 10 µL of sterile 10 mM Tris (pH 8.0). Prior to proceeding with the NEBNext Ultra II DNA library prep kit, 1 µL of DNA was quantified using the Qubit dsDNA HS kit.

**Sequencing.** The concentration of finalized libraries was measured using the NEBNext Library Quant kit for Illumina (NEB #E7630). Samples were pooled to a final concentration of 4 nM per library prior to 50 bp paired-end sequencing on the NovaSeq 6000.

## Data analysis

Read mapping: Raw sequence reads were aligned to the *Drosophila melanogaster* reference genome (UCSC, version dm6) with Bowtie2 in the following command (bowtie2 -N 1 -X 1000 -q -5 0–3 0) [95]. Reads with low mapping quality (MAPQ $\leq$ 10) or redundant reads that mapped to the same location with the same orientation were removed in each library. Overall, there were roughly 5,500,000 reads per library on average.

## Differential analysis

We measured the number of reads in each of the 9 chromatin regions (a total of 42,126 regions) downloaded from the ModENCODE project [75]. The reads were normalized by the library size followed by logarithmic transformation. We identified differential regions by applying a two-sample *t* test and setting the *p*-value cutoff equal to 0.05 and fold-change cutoff to be 1.3-fold. The enrichment of the differential occupancy in each type of the 9 chromatin regions were quantified by a hypergeometric *p*-value using the total number of regions (42,126), the total number of regions in a specific chromatin region, the number of differential regions, and the number of overlapped regions between the differential regions and the chosen chromatin region. To generate the average profile plots, each region from a state was binned into 50 bins, such that the size of bins from different regions can be different. Therefore, the

average plot shows scaled regions. The number of reads located in the bins were counted then normalized by library size and $log_2$ transformation. Last, gaussian smoothing was applied to smooth the average profile.

## Supporting information

**S1 Fig. Heat shock–induced tag switch in transgenic lines.** (**A**) Testis tip of the *nos>FRT-H3A31S-eGFP-FRT- H3A31S-mCherry* and (**B**) *nos>FRT-H3.3S31A-eGFP-FRT-H3.3S31A-mCherry* transgenic lines. Both old histone (eGFP labeled) and new histone (mCherry labeled) signals are shown in GSCs and SGs, 12 hours after heat shock–induced tag switch. Given the cell cycle length for GSCs (12–16 hours) and SGs (10–12 hours) [54], 12 hours post-heat shock recovery time should allow germ cells to be within or just at the accomplishment of the first cell cycle. There were undetectable or low levels of new H3A31S (**A**), but abundant new H3.3S31A (**B**). These cell cycle dependence of each mutant histone is similar to their corresponding incorporation mode as WT H3 is S phase dependent and WT H3.3 is S phase independent, consistent with previous reports [68,29]. GSC, germline stem cell; SG, spermatogonial cell; WT, wild-type.
(PDF)

**S2 Fig. Distribution patterns of old (eGFP labeled) versus new (mCherry labeled) histones in mitotic (prophase) GSCs.** (**A**-**D**) Old versus new control H3 (**A**), mutant H3A31S (**B**), control H3.3 (**C**), mutant H3.3S31A (**D**) distribution in mitotic (prophase) GSCs labeled with Hoechst and Armadillo (white, DNA marker and Hub marker, respectively), showing old (green) and new (red) histone distribution patterns. Asterisk: niche. Scale bars: 5 μm. GSC, germline stem cell.
(PDF)

**S3 Fig. Colocalization of distribution patterns of old versus new histones in prophase or prometaphase SGs and GSCs.** (**A**) Old versus new WT H3 in an SG. (**B**) Quantification colocalization patterns of H3 WT in SGs versus GSCs show significant differences. Unpaired *t* test to compare the 2 individual datasets to each other. *** $P < 0.001$. The data underlying this panel can be found in S8 Table. GSC, germline stem cell; SG, spermatogonial cell; WT, wild-type.
(PDF)

**S4 Fig. Distribution of EdU pulse labeling on the replicative chromatin fibers derived from early-stage germ cells expressing WT H3 versus mutant H3A31S.** Quantification of EdU distribution on H3A31S-labeled chromatin fibers shows significant differences from EdU distribution on WT H3-labeled chromatin fibers. The EdU asymmetry is less on H3A31S-labeled chromatin fibers than on WT H3-labeled chromatin fibers. * $P < 0.05$ using F test to compare variances. The data underlying this panel can be found in S9 Table. WT, wild-type.
(PDF)

**S5 Fig. The old H3.3 and H3.3S31A histones turnover in live GSCs after heat shock.** The time-course recovery experiment was performed using live cell imaging (at 20 hours and 30 hours) after heat shock to examine old histones turnover in GSCs. The 3D quantification of old histone (sum intensity) was performed in GSCs at each time point after heat shock recovery. Individual data points are representative of individual GSCs at each time point. (**A**) H3.3 (20 hours: $n = 11$, 1198965.5 ± 50341.5; 30 hours: $n = 11$, 828469.4 ± 37435.7) and (**B**) H3.3S31A (20 hours: $n = 10$, 1629018.0 ± 107460.6; 30 hours: $n = 10$, 872505.4 ± 86988.8). Error bars represent the SEM. The data underlying these panels can be found in S10 Table.

GSC, germline stem cell; SEM, standard error of the mean.
(PDF)

**S6 Fig. Germ cell–specific expression of the histone transgenes.** Coexpression of the unpaired and eGFP-tagged histone transgenes with the *nanos* promoter produces germ cell and somatic cell tumors where histone-eGFP is expressed only in germ cells. Each of the tagged histones (green) are shown in complementary with somatic cyst cells stained with Traffic Jam (red). Scale bar: 10 μm.
(PDF)

**S7 Fig. Immunoblot of transgenic and endogenous histones in *nos> Upd* testes.** Endogenous and transgenic histones H3 and H3.3 were blotted for in control (C) *nos> Upd* and *nos> Upd; histone-GFP* testes. Endogenous histone H3 and H3.3 (approximately 15 kDa, arrows in respective blots) can be seen in addition to transgenic GFP-tagged histones (approximately 40 kDa, arrow in anti-GFP blot). Ponceau staining of each membrane shows total protein loaded.
(PDF)

**S8 Fig. Average density profiles at transcription start sites in State 1.** Unlike histones H3 and H3A31S (left), the average density plot of histone H3.3 and its mutant H3.3S31A (right) show no differences in occupancy at TSS extracted from chromatin state 1. TSS, transcription start site.
(PDF)

**S1 Table. Quantification for Fig 1 (1E, 1F, and 1H).**
(PDF)

**S2 Table. Quantification for Fig 2 (2B, 2C, and 2E).**
(PDF)

**S3 Table. Quantification for Fig 3F.**
(PDF)

**S4 Table. Quantification for Fig 4E.**
(PDF)

**S5 Table. Quantification for Fig 5E and 5F.**
(PDF)

**S6 Table. Quantification for Fig 6D and 6D'.**
(PDF)

**S7 Table. *P* values for Fig 7C and 7D.**
(PDF)

**S8 Table. Quantification for S3B Fig.**
(PDF)

**S9 Table. Quantification for S4 Fig.**
(PDF)

**S10 Table. Quantification for S5 Fig.** The 3D quantification of old histone [sum intensity (a. u)] was performed in GSCs at each time point (i.e., at 20 hours and 30 hours) after heat shock. Individual data points represent individual GSCs at each time point. GSC, germline stem cell.
(PDF)

**S1 Movie. Related to Fig 3. Asymmetric old and new histone H3 inheritance during asymmetric GSC division.** Live cell imaging of the male *Drosophila* GSC expressing dual color switchable tag (H3-mCherry-EGFP). This video is a reconstruction of individual images (10 × 1-μm interval optical sections per frame). ACD of male GSC showing an asymmetric inheritance of old H3 (green) and new H3 (red) in telophase. The video was acquired at 5-minute intervals for 10–12 hours. Metaphase is used as a landmark to define time point zero, and other time points are labeled as minus minutes prior to metaphase. The quantification is shown in Fig 3. Asterisk: hub. Scale bar: 5 μm. ACD, asymmetric cell division; GSC, germline stem cell.
(AVI)

**S2 Movie. Related to Fig 3. Symmetric old and new histone H3 inheritance during symmetric SG division.** Live cell imaging of the male *Drosophila* SG expressing dual color switchable tag (H3-mCherry-EGFP). This video is a reconstruction of individual images (10 × 1-μm interval optical sections per frame). ACD of male SG showing a symmetric inheritance of old H3 (green) and new H3 (red) in telophase. The video was acquired at 5-minute intervals for 10–12 hours. Metaphase is used as a landmark to define time point zero, and other time points are labeled as minus minutes prior to metaphase. The quantification is shown in Fig 3. Scale bar: 5 μm. ACD, asymmetric cell division; SG, spermatogonial cell.
(AVI)

**S3 Movie. Related to Fig 3. Symmetric inheritance of mutant histone H3A31S during GSC division.** Live cell imaging of the male *Drosophila* GSC expressing dual color switchable tag (H3A31S-EGFP-mCherry-). This video is a reconstruction of individual images (10 × 1-μm interval optical sections per frame). The male GSC division showing a symmetric inheritance of old H3A31S (green) and new H3A31S (red) in telophase. The video was acquired at 5-minute intervals for over 10–12 hours. Metaphase is used as a landmark to define time point zero, and other time points are labeled as minus minutes prior to metaphase. The quantification is shown in Fig 3. Asterisk: hub. Scale bar: 5 μm. GSC, germline stem cell.
(AVI)

**S4 Movie. Related to Fig 3. Symmetric inheritance of histone H3.3 during GSC division.** Live cell imaging of the male *Drosophila* GSC expressing dual color switchable tag (H3.3-EGFP -mCherry). This video is a reconstruction of individual images (10 × 1-μm interval optical sections per frame). ACD of male GSC showing a symmetric inheritance of old H3.3 (green) and new H3.3 (red) in telophase. The video was acquired at 5-minute intervals for over 10–12 hours. Metaphase is used as a landmark to define time point zero, and other time points are labeled as minus minutes prior to metaphase. The quantification is shown in Fig 3. Asterisk: hub. Scale bar: 5 μm. ACD, asymmetric cell division; GSC, germline stem cell.
(AVI)

**S5 Movie. Related to Fig 3. Symmetric inheritance of mutant histone H3.3S31A during GSC division.** Live cell imaging of the male *Drosophila* GSC expressing dual color switchable tag (H3.3S31A-EGFP-mCherry). This video is a reconstruction of individual images (10 × 1-μm interval optical sections per frame). ACD of male GSC showing a symmetric inheritance of old H3.3S31A (green) and new H3.3S31A (red) in telophase. The video was acquired at 5-minute intervals for over 10–12 hours. Metaphase is used as a landmark to define time point zero, and other time points are labeled as minus minutes prior to metaphase. The quantification is shown in Fig 3. Asterisk: hub. Scale bar: 5 μm. ACD, asymmetric cell division; GSC, germline stem cell.
(AVI)

**S1 Raw Images. Original blot and gel images.**
(PDF)

## Acknowledgments

We would like to thank the Chen Lab members for their insightful discussion. We thank Dr. Guangzhe Ge (NHLBI/NIH) for assistance with the ChIC-seq samples.

## Author Contributions

**Conceptualization:** Chinmayi Chandrasekhara, Rajesh Ranjan, Jennifer A. Urban, Brendon E. M. Davis, Xin Chen.

**Data curation:** Jennifer A. Urban, Wai Lim Ku.

**Formal analysis:** Chinmayi Chandrasekhara, Rajesh Ranjan, Jennifer A. Urban, Brendon E. M. Davis, Wai Lim Ku, Jonathan Snedeker.

**Funding acquisition:** Jennifer A. Urban, Keji Zhao, Xin Chen.

**Investigation:** Chinmayi Chandrasekhara, Rajesh Ranjan, Jennifer A. Urban, Brendon E. M. Davis, Wai Lim Ku, Xin Chen.

**Methodology:** Rajesh Ranjan, Jennifer A. Urban.

**Project administration:** Xin Chen.

**Resources:** Chinmayi Chandrasekhara, Rajesh Ranjan, Jennifer A. Urban, Wai Lim Ku.

**Software:** Wai Lim Ku.

**Supervision:** Keji Zhao, Xin Chen.

**Validation:** Chinmayi Chandrasekhara, Rajesh Ranjan, Jennifer A. Urban.

**Visualization:** Chinmayi Chandrasekhara, Rajesh Ranjan, Jennifer A. Urban, Jonathan Snedeker.

**Writing – original draft:** Chinmayi Chandrasekhara, Xin Chen.

**Writing – review & editing:** Chinmayi Chandrasekhara, Rajesh Ranjan, Jennifer A. Urban, Brendon E. M. Davis, Keji Zhao, Xin Chen.

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
