## [Editor Report · Decision Letter 0]

7 Sep 2022

Dear Dr Chen, 

Thank you for submitting your manuscript entitled "Roles of the distinct N-terminal amino acid between H3 and H3.3 in Drosophila male germline stem cell lineage" for consideration as a Short Report by PLOS Biology.

Your manuscript has now been evaluated by the PLOS Biology editorial staff as well as by an academic editor with relevant expertise and I am writing to let you know that we would like to send your submission out for external peer review. I should note, that at this stage we have yet to make a firm call about whether your manuscript goes far enough for PLOS Biology, without, for example, providing more details into the functional significance of this phenomenon. We will therefore be looking for enthusiasm from reviewers regarding its fit for our Short Report format. 

Before we can send your manuscript to reviewers, we need you to complete your submission by providing the metadata that is required for full assessment. To this end, please login to Editorial Manager where you will find the paper in the 'Submissions Needing Revisions' folder on your homepage. Please click 'Revise Submission' from the Action Links and complete all additional questions in the submission questionnaire.

Once your full submission is complete, your paper will undergo a series of checks in preparation for peer review. After your manuscript has passed the checks it will be sent out for review. To provide the metadata for your submission, please Login to Editorial Manager (https://www.editorialmanager.com/pbiology) within two working days, i.e. by Sep 09 2022 11:59PM.

Kind regards,

Lucas

Lucas Smith, Ph.D.

Associate Editor

PLOS Biology

lsmith@plos.org

---

## [Decision Letter · Decision Letter 1]

18 Oct 2022

Dear Xin,

Thank you for your patience while your manuscript "Roles of the distinct N-terminal amino acid between H3 and H3.3 in Drosophila male germline stem cell lineage" was peer-reviewed at PLOS Biology. It has now been evaluated by the PLOS Biology editors, an Academic Editor with relevant expertise, and by several independent reviewers. 

The reviews of your manuscript are appended below. As you will see from their comments, while the reviewers agree that the findings presented here are interesting, they also feel additional work is needed to better support and explain the conclusions of your study. Given the reviewer concerns, we cannot accept the manuscript in its current form, but would welcome a revised manuscript that thoroughly addresses the reviewer comments.

Given the extent of revision needed, we cannot make a decision about publication until we have seen the revised manuscript and your response to the reviewers' comments. Your revised manuscript is likely to be sent for further evaluation by all or a subset of the reviewers.

**IMPORTANT - SUBMITTING YOUR REVISION**

*Re-submission Checklist*

*Published Peer Review*

*PLOS Data Policy*

*Blot and Gel Data Policy*

Sincerely,

Lucas

Lucas Smith, Ph.D.

Associate Editor

PLOS Biology

lsmith@plos.org

REVIEWS:

Reviewer #1: This study uses the powerful Drosophila male germline stem cell (GSC) system to probe the function of the 31 residue of canonical replicative histone H3 and the histone H3 variant H3.3. Intriguingly, this is the only residue that differs between the N terminus of H3 and H3.3. The authors strategy is to (over) express GFP-tagged versions of wild type H3 or mutant H3A31S, and wild type H3.3 or H3.3S31A in GSCs and early germ cells. They subsequently monitor effects on GSC maintenance, testes development and fertility. In addition, using their previously published genetic system that allows pre-existing and newly synthesised versions of proteins to be distinguished, the authors track the fate of new and old wild type or mutant histones during asymmetry cell division (ACD). The goal is to determine whether residue 31 functions in the asymmetric distribution of histones that has been shown to occur in GSC and daughter (gonialblast; GB) cells. This study is novel and interesting in that it addresses the function of amino acid 31 of H3/H3.3 in the context of a developing tissue. The genetic switch assays utilised, combined with live imaging in testes, are sophisticated and cutting edge. I would recommend this report for publication after addressing the following 3 major points: 

1. One general query I have relates to the expression level of wild type H3 or mutant H3A31S, and wild type H3.3 or H3.3S31A in the different experiments. For example, was the expression level comparable between H3 and H3A31S? The authors should at least provide some estimation of the level of GFP-tagged wild type/mutant histone relative to endogenous H3 expression. Also in Figure 7 while the incorporation pattern of the tagged, over-expressed versions of H3/H3.3 are considered, were any changes to endogenous H3 pattern noted upon transgene expression? 

2. In Figure 3, the authors show that old H3A31S displays less asymmetry compared to wild type old H3, concluding that Ala31 is required for asymmetry. This observation is further supported by data presented in Figure 5 in which parental H3K27me3 redistribution is assessed. According to the quantitation, the H3K27me3 distribution pattern is disrupted. However, it is perhaps a leap to make the general conclusion that at the replication fork is disrupted, at least without validation by an independent method. For example, it is possible that the pattern is disrupted as H3K27 methylation status is affected by the H3A31S mutation or its over expression. For this reason the claim the old histone recycling is disrupted should be toned down. Also in this Figure, can the authors explain why EdU coats only one stand in 5A, but both strands in 5B? Finally, PCNA staining should be included in 5B. 

3. In Figure 3, the authors show that old H3.3S31A is symmetrically distributed, comparable to wild type H3.3, concluding Ser31 is not critical for asymmetry. This observation is further supported by the claim in Figure 6 that H3.3S31A turnover is faster than that of wild type H3.3. However, an alternative explanation is that H3.3S31A is more sensitive to fixation. Live snapshot should be quantified to exclude this possibility. 

Other minor points:

In Figure 1, the authors convincingly show that compared to expression of H3, H3A31S expression in testes leads to an over-population of early germ cells (including GSCs). What is not clear is whether over-expression of H3 alone can drive this effect i.e. what is the basal number of GSCs in the control line? It is also not clear why only 33% of testes analysed showed this phenotype (Fig 1E). The authors should provide an explanation. 

In Figure 2, the authors convincingly show that compared to expression of H3.3, H3.3S31A expression in testes leads to a gradual loss of GSCs that correlate well with a decline in fertility over time. Notably, even in the line expressing H3.3 wild type, GSCs were reduced after 10 days. My guess is that this is due to ageing, but the authors should address this point. Figure 2D and 2E show a reduced level of Stat92E present in H3.3S31A GSCs. For the quantitation in 2E, it appears that some GSCs showed no Stat92E signal. Perhaps the authors can expand on whether a reduction in Stat92E signal is sufficient to alter GSC identity or whether a total loss is required? 

In Figure 4E, the calculated Spearman's rank correlation for H3A31S and H3.3S31A appear similar. Yet from the images presented in 4E, the degree of overlap does not appear that similar. Are the values for H3A31S and H3.3S31A significantly different? 

Typo Figure 7A 'targeting' 

Typo Figure 1 legend B 'in testes' 

Reviewer #2: In this work, Chandrasekhara et al., use the Drosophila male germline stem cell system to study the molecular mechanism determining the distribution pattern of new vs old histones. Taking the advantage of different incorporation patterns between histone H3 and H3.3 (with the former being asymmetric and the latter randomly incorporating between two sister chromatids), they identified the difference of the 31st amino acid (A in H3 and S in H3.3) is critical for determining different segregation patterns towards two sister chromatids in germ cells. By swapping the 31st amino between these histones, they found that H3's 31st A residue is critical for its preferential incorporation toward one of the sisters during S phase, whereas H3.3's 31st S is critical for regulating turnover time of H3.3 to continuously replace preexisting histones. Moreover, they show that one of the replicating strands which is likely incorporating old histones contains more H3K27me3 marks. Furthermore, they conducted ChIC seq to show where these ectopically expressed histones H3, H3.3 and their swapped versions bind in the genome.

These findings are certainly of interest to the broad readership of Plos Biology. However, some results presented here are very difficult to interpret, especially the ChIC seq. I would suggest to clarify these uncertain points for better understanding of what happening in the cells.

Main points;

1) The authors show that H3.3S31A and H3A31S both show phenotypes possibly due to altered incorporation patterns of new/old molecules toward two sister chromatids. Do these swapped versions still keep same replication dependency/independency? 

2) Based on a previous report from same group (https://www.ncbi.nlm.nih.gov/pmc/articles/PMC6684448/), asymmetric incorporation of H3 globally to the entire length of a sister chromosome depends on the unidirectional replication mechanism. Does H3A31S disrupt unidirectional replication or do the authors think another mechanism is at play?

3) In the Fig4 colocalization assay, do the authors consider H3 and H3.3 visualized here are all on DNA? Do these images show any fraction in the nucleoplasm? Colocalization with DNA may help to interpret the data.

4) Same figure (Fig4). Do the authors think the distinct localization of old/new H3 occurs in any germ cell or only GSCs? 

5) Same figure again (Fig4). Do the authors still see a distinct localization pattern in H3T3A? If it was reported previously, please describe.

6) Why does Figure5 show Edu in a single strand on the top and both strands on the bottom? Does this mean the DNA synthesis pattern is different between them? A couple of typos here; PCNA, H3 WT in B'.

7) ChIC seq—do Upd-induced tumor cells incorporate new/old H3 differently toward two sisters? Any previous report or data is helpful.

8) ChIC seq—Why does H3 show distinct peaks for these regions? Is there any effect of endogenous H3? It will be helpful if H3 or H3.3's general ChIP seq pattern is compared with their data is explained.

Minor:

Where whole mount vs squash methods used?

Reviewer #3: This manuscript reports a role for the amino acid at position 31 of histones H3 and H3.3 in inheritance of H3 during the asymmetric cell division of Drosophila male germline line stem cells (GSCs). Previous studies by this group have established that during this asymmetric cell division histones H3 is preferentially inherited by the GSC while histone H3.3 is not. This is a fascinating problem with important implications for the role of histone inheritance in acquisition of cell identity during asymmetric cell divisions. This study now identified the sequence difference between H3 and H3.3 that dictate their distinct inheritance patterns and also reports on the global localization patterns of H3 and H3.3 in GSC-like cells. The authors report the following main findings. They construct inducible H3A31S and H3.3S31A mutant histone lines and using these reagents they observe that early-stage germ cells are over-populated in the H3A31S-expressing testes while there is a loss of germ cells in testes expressing H3.3S31A. They further show that asymmetric H3 inheritance is disrupted in the H3A31S-expressing GSCs due to mis-incorporation of old histones between sister chromatids during DNA replication, H3.3S31A mutation leads to increased old histone turnover in the GSCs, and using a chromatin immune-cleavage assay show that H3A31S has enhanced occupancy at promoters of active genes while H3.3S31A is more enriched at transcriptionally silent intergenic regions compared to H3.3. From a technical point of view, the authors provide old and new H3 and H3.3 localization data based on new live cell image and 3D reconstruction that confirms their previous findings based on fixed cell imaging data. The results indicate an important new role for the N termini of histone H3 and H3.3 in the regulation of their localization and asymmetric histone inheritance and will be of great interest to the field. The manuscript is in principle suitable for publication in PlosBiology after the authors address the following minor points regarding data presentation.

1. The authors should describe in the text and Figures 1 and 2 legends how they are identifying GSCs and other cell types. The location of the niche and the hub cell indicated by asterisks in various figures and what various staining events indicate should be explained to allow a general audience to follow the results.

2. The effects that the authors show in Figure 3 (panels A-D) look small and at least visually the asymmetric inheritance of H3 in GSCs doesn't look as impressive as what the authors have reported previously or show by quantification of live cell images in panel F. Perhaps the authors can present 3D image reconstructions (as collapsed optical stacks) of relevant GSCs in panels A to D that better show the asymmetric inheritance and its loss in H3A31S etc.

3. In Figure 4, the S10P signal for mitotic chromosomes seems asymmetric in the GSCs. This seems unexpected. Can the authors explain this observation? Would it make sense to normalize the GFP and red signal to the H3S10P signal?

4. In Figure 5, the EdU signal seems asymmetric in the chromatin fibers shown in panel A (WT H3) but is symmetrical for the fiber example shown in panel B (H3A31S testes). Can the author provide an explanation for this difference? Also, the legend should describe the reasoning for staining for PCNA (I assume as an indicator of the lagging strand), EdU, etc.

---

## [Decision Letter · Decision Letter 2]

8 Mar 2023

Dear Xin,

Thank you for your patience while we considered your revised manuscript "Roles of the distinct N-terminal amino acid between H3 and H3.3 in Drosophila male germline stem cell lineage" for publication as a Short Report at PLOS Biology. This revised version of your manuscript has been evaluated by the PLOS Biology editors, the Academic Editor and the original reviewers.

The reviews are appended below. As you will see, the reviewers are largely satisfied by the revision, although Reviewer 2 has highlighted a point that needs additional consideration and further discussion in the manuscript. Based on the reviews, we are likely to accept this manuscript for publication, provided you satisfactorily address the remaining points raised by reviewer 2. Please also make sure to address the following data and other policy-related requests.

**EDITORIAL REQUESTS: 

1) TITLE: After some discussion, we think the title should be edited slightly for clarity. If you agree we would suggest that you change it to something like "A single N-terminal amino acid determines the distinct roles of histones H3 and H3.3 in the Drosophila male germline stem cell lineage". 

2) ARTICLE TYPE: While your manuscript was originally submitted as a Short Report, we think that it would be better suited for our Research Article format, as Short Reports cannot have more than 4 figures. We therefore ask that you change the article type to Research Article. 

3) DATA AVAILABILITY: Thank you for depositing your ChIC-seq data to the GEO repository. I could not seem to find a reviewer token to access the data (sorry if I missed it). Can you please provide me with one so I can check the data meets our requirements?

Also, thank you for providing the numerical data underlying your figures in supplementary tables. For the most part, this data meets our requirements, however I did not see tables related to Fig S3B or Fig S4. Can you please make sure to provide tables containing the underlying data plotted in these figures?

Please also be sure to cite this underlying data in each relevant figure legend. For example, to each figure legend you can add the sentence "the data underlying this figure can be found in Table S___". 

4) BLOT REPORTING: We require the original, uncropped and minimally adjusted images supporting all blot and gel results reported in an article's figures or Supporting Information files. We will require these files before a manuscript can be accepted so please prepare and upload them now. Please carefully read our guidelines for how to prepare and upload this data: https://journals.plos.org/plosbiology/s/figures#loc-blot-and-gel-reporting-requirements

Looking at Figure S7, it seems these blots are slightly cropped. Can you provide the full, unadjusted and uncropped blots accompanying this figure (annotated as described in our guidelines above)? 

5) METHODS: I noticed that the methods section for your manuscript is currently contained in a supplemental file. Can you please move this into the main text?

We expect to receive your revised manuscript within two weeks. 

*Published Peer Review History*

*Press*

Sincerely,

Lucas

Lucas Smith, Ph.D.

Associate Editor,

lsmith@plos.org,

PLOS Biology

Reviewer remarks:

Reviewer #1: In this revised version of the manuscript, the authors have addressed all my comments. 

In particular, the addition of the H4K20me3 data to Figure 5, as well as the quantitation presented in Figure S4 are important and interesting results that broaden the scope of the findings. 

The western blots present in Figure S7 show no major or obvious effects of transgene expression on endogenous histone level and I appreciate that cell-specific protein extractions from testes are not technically possible. 

Reviewer #2: The manuscript has been improved with several new figures. Based on the information in authors' response, there are significant delay of lagging strand synthesis in GSCs, which authors plan to report in their future publication. If this is the case, newly formed histones (color switched to red) will of course distribute to lagging strand based on red protein availablity at the time of incorporation. However, old histones in original color does not necessarily "preexisting" histones. In this case, histone turn over time will be more important factor to understand distribution of each histone species (old vs new, preexisting vs newly formed). This also affect the understanding of chic seq distribution data. In fact, endogenous histones are expressed specifically in S-phase, but UAS based expression timing must be continuous. I would suggest to discuss potential reason and effect of asymmetric EdU strand observation and add more rigorous discussion about histone turnover timing.

Reviewer #3: The authors have addressed my concerns satisfactorily and I support publication.

---

## [Editor Report · Decision Letter 3]

29 Mar 2023

Dear Xin,

Thank you for the submission of your revised Research Article "A single N-terminal amino acid determines the distinct roles of histones H3 and H3.3 in the Drosophila male germline stem cell lineage " for publication in PLOS Biology. On behalf of my colleagues and the Academic Editor, Yukiko Yamashita, I am pleased to say that we are satisfied by the changes made in this revision and we can in principle accept your manuscript for publication. 

As a note, before your manuscript can be published, you will be asked to address any formatting and reporting issues. These will be detailed in an email you should receive within 2-3 business days from our colleagues in the journal operations team; no action is required from you until then. Please note that we will not be able to formally accept your manuscript and schedule it for publication until you have completed any requested changes.

PRESS

Sincerely, 

Luke

Lucas Smith, Ph.D.

Associate Editor

PLOS Biology

lsmith@plos.org